

# Scaling of natural fracture patterns at Swift anticline, NW Montana: the influence of structural position, lithology, and observation scale

Adam J. Cawood[1,a], Hannah Watkins[1], Clare E. Bond[1], Marian J. Warren[2], Mark A. Cooper[1,3]

[1] School of Geosciences, University of Aberdeen, King's College, Aberdeen AB24 3UE, UK
[2] Jenner GeoConsulting Inc., 107 Lake Tahoe Place SE, Calgary, Alberta, T2J 4B7, Canada
[3]Sherwood GeoConsulting Inc., 140 Lake Mead Cr SE, Calgary, Alberta, T2J 4A1, Canada
[a] Present address: Southwest Research Institute, 6220 Culebra Rd, San Antonio, Texas 78238-5166, USA

*Correspondence to*: Adam J. Cawood (adam.cawood@swri.org)

**Abstract.** Natural fracture patterns have long been associated with fold formation. Conceptual models of fold associated fractures are used to predict fracture networks and hence subsurface properties such as fracture connectivity, intensity and fluid flow. Subsurface datasets typically lack the resolution or coverage to adequately sample fracture networks in 3D, however, and geometric properties are typically extrapolated from available data (e.g., seismic data or wellbore image logs).

Here we assess the applicability of extrapolating fracture properties (orientation, length and intensity) from one observation scale to another in a structurally complex setting and assess the interplay of fracture scaling with geological controls on fracture development. Fracture patterns are investigated at an outcrop exposure of layered carbonate rocks at Swift anticline, NW Montana. Data derived from high-resolution field images, medium resolution digital outcrop data, and relatively low resolution satellite imagery are leveraged to (i) assess interacting structural and stratigraphic controls on fracture development, and (ii)

compare estimated fracture properties derived from multiple observation scales. We show that hinge-parallel and hinge-perpendicular fractures (i) make up the majority of fractures at the site, (ii) are consistently oriented with respect to the fold hinge and, (iii) exhibit systematic increases in intensity towards the anticline hinge. These fractures are interpreted as having formed during folding. Other fractures recorded at the site exhibit inconsistent orientations, show no systematic trends in fracture intensity, and are interpreted as unrelated to fold formation. Fracture orientation data exhibit greatest agreement across

observation scales at hinge and forelimb positions where hinge-parallel and hinge-perpendicular fracture sets are well developed, and little agreement on the anticline backlimb, where fracture orientations are less predictable and more dispersed. This indicates that the scaling of fracture properties at Swift anticline is spatially variable and partly dependent on structural position. Our results suggest that accurate prediction and extrapolation of natural fracture properties in contractional settings requires assessment of structural position, lithologic variability, and spatially variable fracture scaling relationships, as well as

consideration of deformation history before and after folding.



## 1 Introduction

The ability to predict accurately natural fracture attributes (e.g., aperture, length, orientation) and patterns (e.g., density, connectivity) has implications for resource management and waste disposal in the subsurface. Natural fractures typically enhance the porosity and permeability of subsurface rock volumes and predicting fracture attributes is therefore important for

a range of activities related to subsurface fluid flow regimes. Specific applications include $CO_2$ sequestration (e.g., Iding and Ringrose, 2010; Bond et al., 2013, 2017; Gholami et al., 2021; Kou et al., 2021), hazardous waste disposal (e.g., Green and Mair, 1983; Gautschi, 2001; Morris et al., 2004; Yu et al., 2021; Ishii, 2022), groundwater management (e.g., Streltsova, 1976; Bachu, 1995; Ferrill et al., 1999; Medici et al., 2021; Moore and Walsh, 2021), hydrocarbon extraction (e.g., Thomas et al., 1983; Mäkel, 2007; Rawnsley et al., 2007; Li and Lee, 2008; Spence et al., 2014; Gong et al., 2021) and geothermal energy

production (e.g., Bödvarsson and Tsang, 1982; Watanabe and Takajashi, 1995; Shaik et al., 2011; Fox et al., 2013; Glaas et al., 2021; Chabani et al., 2022). Despite the range of applications that rely on knowledge of subsurface fracture properties, accurate fracture prediction remains challenging due to (i) the spatial variability and complexity of natural fracture networks, and (ii) difficulties related to sampling fracture populations in subsurface datasets.

Subsurface data (e.g., wellbore information and seismic imaging) provide constraints on fracture properties, but limits to

the coverage and resolution of these datasets often result in highly uncertain predictions of fracture properties at depth. Wellbore data can provide direct, in-situ fracture measurements (e.g., orientation data from image logs) which can be used for generating predictive fracture models (e.g., Cooper, 1991; Aliverti et al., 2003; Nadimi et al., 2020). Wellbores are generally widely spaced in the subsurface, however, and predictions from well data typically suffer from sampling biases (e.g., Sun et al., 2016; Yin and Chen, 2020). Extrapolating fracture properties (e.g., length, orientation, abundance) away from wells or

interpolating between wells is therefore not straightforward, with predictions prone to substantial uncertainties (e.g., De Marsily, 2005; Ma et al., 2007). Remote sensing (e.g., seismic reflection) data provide a potential alternative for sampling subsurface fracture populations but these data typically lack the resolution to image all but the largest fractures or discontinuities in the subsurface (e.g., Marrett and Allmendinger, 1992; Yielding et al., 1996; Rawnsley et al., 2007; Worthington & Lubbe, 2007; Dimmen et al., 2023). As such, seismic data are generally more useful for providing contextual

information (e.g., structural position, distance to major faults) than for directly imaging fracture networks in detail.

Faced with limited information about subsurface fracture properties, geoscientists may supplement subsurface datasets with information derived from appropriate outcrop analogues (e.g., Inks et al., 2015; Becker et al., 2018; Ukar et al., 2019). Recent advances in digital photogrammetry and digital mapping approaches (e.g., James and Robson, 2012; Cawood et al., 2017, 2022; Corradetti et al., 2018; Bowness et al., 2022) provide the opportunity to map and measure fractures at outcrop

across a range of spatial scales (e.g., Strijker et al., 2012; Seers and Hodgetts, 2014; Hardebol et al., 2015). By integrating traditional fieldwork with modern digital approaches, fracture characterization at outcrop can potentially (i) address sampling gaps in subsurface datasets (Fig. 1), and (ii) be leveraged to generate multi-scale predictions of inherently heterogeneous fracture populations (e.g., Vollgger and Cruden, 2016; Smeraglia et al., 2021).





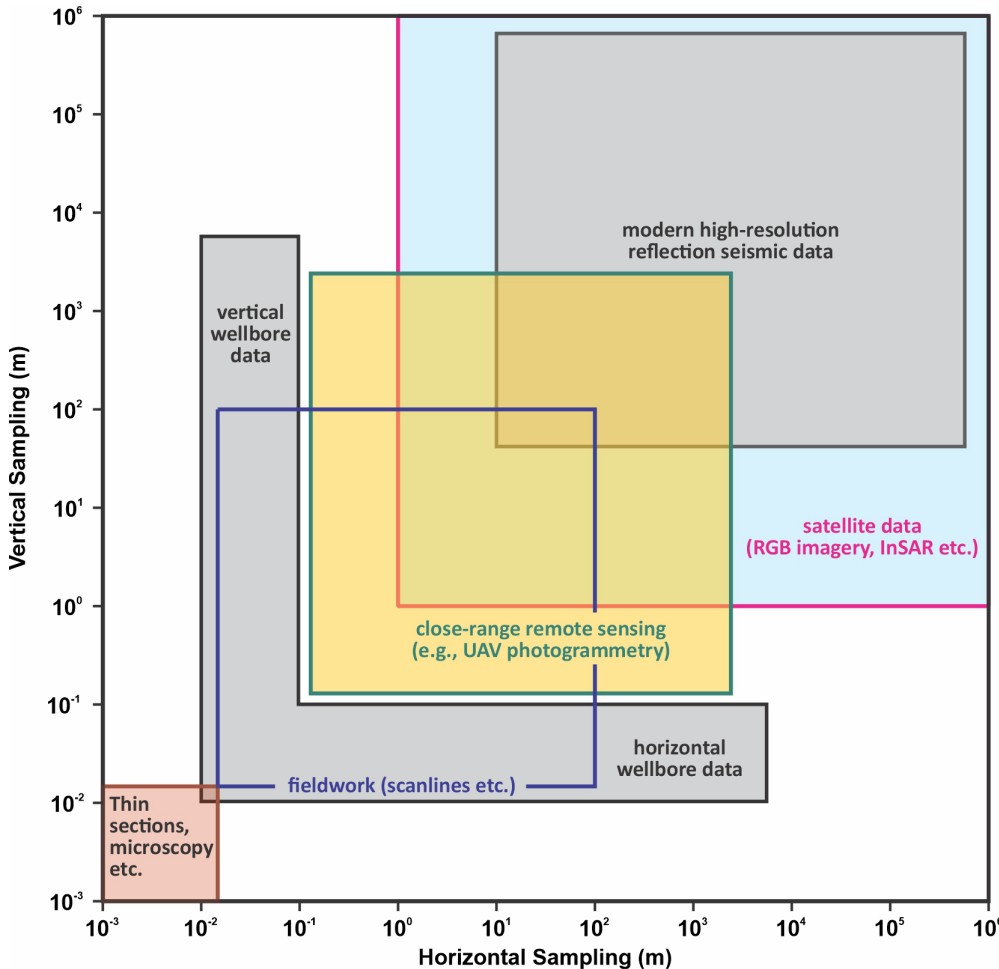

**Figure 1: Scale of geological datasets at which natural fracture networks are typically characterized. Filled grey boxes represent subsurface datasets; unfilled boxes are other data types. The approximate sampling scale for fieldwork refers to the scale at which fracture patterns can be comprehensively sampled using traditional field methods such as fracture scanlines or sampling windows.**

The controls on fracture properties in folded sedimentary rocks have been investigated by numerous workers. Early conceptual models of fracture development predict the presence of discrete, systematic fracture sets on contractional anticlines (Fig. 2), where fracture orientations are consistent with the orientation of the fold on which they occur (e.g., Price, 1966; Stearns, 1964, 1969; Stearns and Friedman, 1972; Hancock, 1985). Subsequent studies have shown that these relatively simple conceptual relationships may be modified by a range of lithological, mechanical, and structural factors (e.g., Cosgrove and Ameen, 1999; Cooper et al., 2006; Wennberg et al., 2007; Bergbauer and Pollard, 2004; Watkins et al., 2015, 2018; Awdal et al., 2016). Documented lithological influences on fracture formation include rock competence (e.g., McGinnis et al., 2017; Bowness et al., 2022), grain size or porosity within units (e.g., Hanks et al., 1997; Wennberg et al., 2006), mechanical layer thickness (e.g., Ladeira and Price, 1981; Narr and Suppe, 1991; Wu and Pollard, 1995), and bed interface characteristics (e.g., Cooke and Underwood, 2001; Cooke et al., 2006; McGinnis et al., 2017), among other factors.



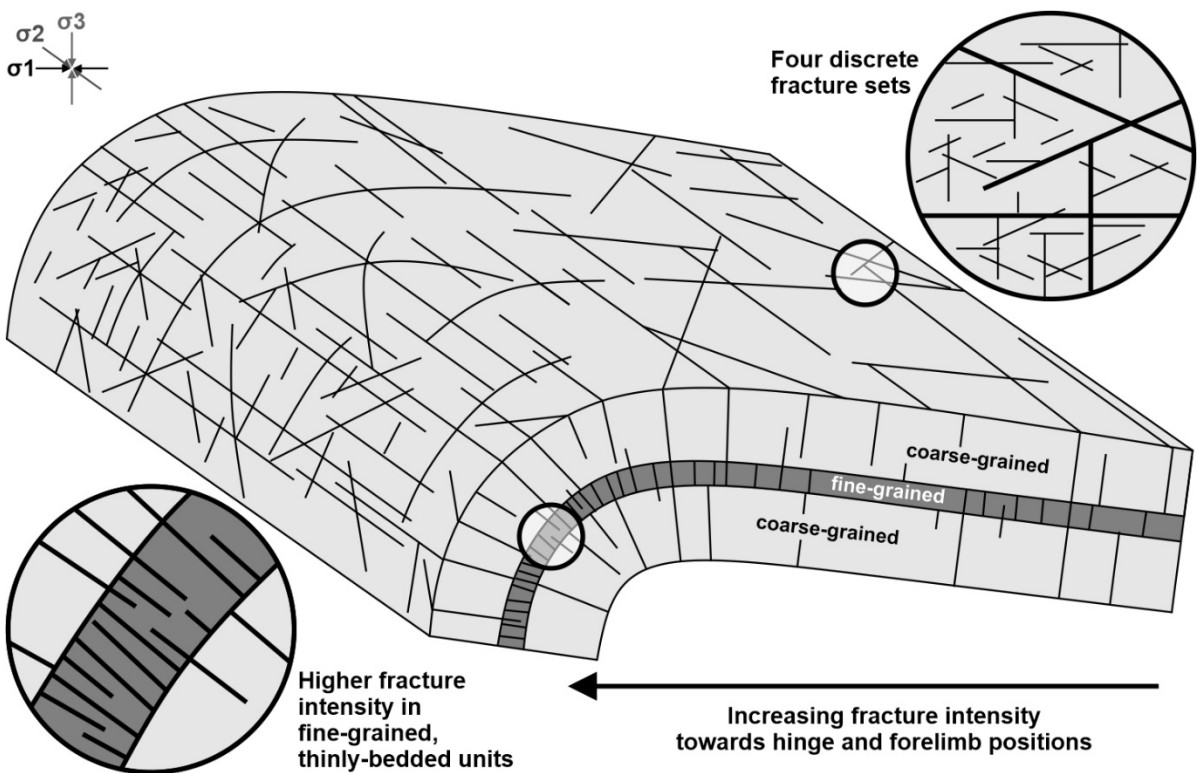

**Figure 2: Conceptual diagram showing established relationships between geological properties and fracture attributes in folded sedimentary rocks. Depicted relationships are: (i) increased fracture intensities at hinge proximal or high curvature zones, (ii) higher fracture intensities in fine-grained or thinly bedded carbonate lithologies, and (iii) the presence of four discrete fracture sets on contractional anticlines. Based on conceptual models by several authors (e.g., Price, 1966; Stearns, 1964, 1969; Stearns and Friedman, 1972; Hancock, 1985; Watkins et al., 2015, 2019)**

Structural controls on fracture attributes include proximity to faults (e.g., Caine et al., 1996; Tamagawa & Pollard, 2008; McGinnis et al., 2015), structural position on folds (e.g., Harris et al., 1960; Hennings et al., 2000; Watkins et al., 2015, 2018) and fold curvature in both dip and strike directions (e.g., Lisle 1992, 1994; Fischer and Wilkerson, 2000). Regional or local stresses and stress perturbations (e.g., Hancock 1985; Tamagawa and Pollard, 2008; Ferrill et al., 1999), burial history and progressive diagenesis (e.g., Laubach et al., 2009; Hooker et al., 2013), and previous episodes of deformation (e.g., Agosta et al., 2010; Casini et al., 2011; Ferrill et al., 2021) are among some of the other factors that may influence fracture network properties. Each of the relationships outlined above may impart spatial variability to natural fracture networks and as a result, fracture properties may vary both in 3D and across spatial scales (e.g., Gillespie et al., 1993, 2001; Castaing et al., 1996; Odling, 1997; Bonnet et al., 2001; Bossennec et al., 2021).

Here we combine 3D photogrammetric reconstruction techniques with field-based measurements and Google Earth imagery to perform a multiscale assessment of fracture properties at Swift anticline, NW Montana. We assess (i) the link between lithology (grain size, rock texture) and fracture intensity, (ii) the influence of structural position vs. fracture orientations and fracture intensity, and (iii) the effects of observation scale on estimated fracture properties. By characterizing



structural and stratigraphic controls on fracture development at multiple observation scales, we provide insights into the scale dependence of fracture formation in deformed multilayer systems, with implications for predicting fracture network properties in the subsurface.

## 2 Geological Setting

The Sawtooth Range is a NNW-SSE trending fold-thrust belt that marks the eastern edge of the Rocky Mountains in NW

Montana (Fig. 3A, B). Cambrian through Cretaceous stratigraphy is deformed and exposed in the Sawtooth Range (Fig. 3C, D). This belt of exposed thrusts and related folds is bound to the west by the Lewis-Eldorado Thrust system and to the east by Jurassic-Paleogene foreland basin deposits associated with the Cordilleran Orogeny (Fuentes et al., 2012). The main phase of fold-thrust deformation in the Sawtooth Range is interpreted to have occurred during late Cretaceous to Palaeocene (Fuentes et al., 2012). Thrusts within the Sawtooth Range are generally closely-spaced, laterally continuous, and westward dipping, and

exhibit a general trend for increased dips westwards, towards the hinterland (Fig. 3C, D). The Sawtooth Range is interpreted as a thin-skinned deformation belt (Mudge, 1982; Mitra, 1986; Holl & Anastasio, 1992; Fuentes et al., 2012) and the stacked thrust sheets of the Sawtooth Range have been interpreted as an exhumed and eroded thrust duplex that formed below the overlying Lewis-Eldorado thrust (Ward & Sears, 2007).

Several studies have focused on fracture patterns within Mississippian carbonate rocks at localities in the Sawtooth

Range. Early work by Stearns (1964, 1969) and by Stearns and Friedman (1972) focused on fracture orientations at Teton anticline (c. 35 km to the south of Swift anticline). This work led to the development of strategies for differentiating between shear vs. extension fractures on anticlines based on their orientations with respect to the fold hinge. The results of these studies led to the widespread use of general models for predicting fracture orientations on and around open folds (e.g., McQuillan, 1973; Fisher & Wilkerson, 2000; Cooper et al., 2006). Later work at Teton anticline focused on fracture spacing (Sinclair,

1980) and the effects of curvature (Spooner, 1984) and structural evolution (Ghosh & Mitra, 2009; Burberry et al., 2019) on fracture attributes. Studies at Swift anticline have related fracture properties at the site to a range of geological factors, including extension driven by flexural loading (Ward & Sears, 2007), variable lithological properties in exposed units (Watkins et al., 2019), and regional stress rotations (Singdahlsen, 1986). Swift anticline has also been used as a direct surface analogue for subsurface gas fields in the eastern Rockies at Waterton, southern Alberta, Canada (Rawnsley et al., 2007).

## 3 Study Area

Swift anticline lies at the eastern edge of Swift Reservoir, NW Montana (Fig. 4A). The present-day erosion level across the anticline exposes carbonates of the Dupuyer Creek Unit (Nichols, 1984, 1986), of the upper part of the Mississippian Castle Reef Formation (Madison Group). At isolated localities, unconformably overlying fine-grained clastic rocks of the Jurassic Ellis Group are preserved (e.g., Fig. 4B). The Mississippian to Jurassic unconformity is widespread across NW Montana &





SW Alberta, and records non-deposition and/or erosion on a possible forebulge before initial deposition in the Cordilleran foreland basin (Ward and Sears, 2007; Fuentes et al., 2012). The Dupuyer Creek Unit makes up most of the exposed strata at Swift anticline and records multiple cycles of carbonate deposition in a shallow water environment, from high-energy, open marine conditions to a tidally influenced interior ramp setting (Mudge, 1982). Strata within the Dupuyer Creek Unit display significant variability in both composition and texture (Watkins et al., 2019), as defined by cyclical variations in depositional

environment (e.g., Nichols, 1984).

**Figure 3: Regional and geological context for Swift Anticline. (A) Regional overview map showing the location of the Sawtooth Range in NW Montana. Generated from satellite imagery (© Google Earth/Landsat/Copernicus) and regional elevation data (ASTER**
**GDEM). (B) Enlarged map of Montana showing location of the Sawtooth Range and simplified structural configuration of the area, modified from Mudge (1982). (C) Simplified geological map for the central part of the Sawtooth Range, modified from Mudge (1982), Mudge & Earhart (1983), and Watkins et al. (2019). (D) Cross-section across Swift Anticline and surrounding area showing general structural geometries, modified from Watkins et al. (2019).**





**Figure 4: Multi-scale imagery of Swift Anticline. (A) Satellite image (© Google Earth/Landsat/Copernicus) showing large-scale, vegetated fractures on the crest of the structure. Ground pixel resolution = ca. 0.35 m. Annotations show locations and look directions for B, C, D, and approximate structural positions on the fold. (B) UAV acquired aerial image of Swift Anticline, looking to SSE along the crest of the structure. Swift Reservoir spillway, exposed fold forelimb and sub-Jurassic unconformity in the foreground of the image. Mm = Mississippian Madison Group; Je = Jurassic Ellis Group. (C) UAV acquired aerial image of the anticline, looking NNW along the crest of the structure. The stepped erosional profile along the anticline crest allows lithological boundaries to be mapped across the structure. (D) Field image of highly fractured coral boundstone unit exposed near the dam spillway.**

Swift anticline is situated in the footwall of an imbricate stack of thrust sheets involving primarily Cambrian to Devonian strata at outcrop (Fig. 3C). The fold is interpreted as a hanging-wall anticline above an ENE-verging thrust fault (Watkins et al., 2019), and is marked by a tight fold hinge with a narrow hinge zone, and steeply dipping to overturned beds in the forelimb of the structure (Fig. 4B). The anticline trends NNW-SSE and is characterised by an arcuate axial trace, which records some variation in its orientation along the crest of the structure (Fig. 4A). The stepped erosional profile across the crest of the



structure (Fig. 4C) exposes several stratigraphic levels within the Dupuyer Creek Unit; the current erosion surface also includes a number of well exposed, areally extensive fractured bedding surfaces (e.g., Fig. 4D) at multiple along-strike locations and forelimb, hinge and backlimb positions. This extensive exposure of fractured bedding surfaces makes Swift anticline a suitable site to examine, at a range of scales, the link between folding and fracturing in multi-layered carbonate stratigraphy.

## 4 Data and methods

Bedding, fault and fracture orientation measurements were collected at the study site using handheld analogue (Silva) and digital (FieldMove on iPad) compass clinometers. These data were used to characterize overall structural geometries at the site and to ground-truth digitally-derived fracture orientations and supplement remotely acquired data. Sedimentary logging was carried out to capture variations grain size and rock texture through the exposed section. In addition to field-based measurements and observations, digital imagery was acquired at multiple scales at the site to assess scale dependent variations in fracture attributes.

Digital imagery and associated data at three observation scales were used for fracture characterization:

1. Satellite imagery (Google Earth, 2018) provided a large-scale, lower-resolution dataset, with an estimated ground-pixel resolution of 0.35 m over the study area. This imagery was used for preliminary digital mapping and generation of a large-scale, low-resolution fracture map.

2. 3D photogrammetric reconstruction of the study site was achieved through acquisition of low-altitude aerial imagery across the structure. 22 manually piloted UAV flights yielded 2987 aerial images, acquired at a range of altitudes (5-97 m) above the outcrop surface. Digital photogrammetric processing was carried out using Agisoft Photoscan Professional 1.6, according to established protocols (e.g., Bemis et al., 2014; Cawood et al., 2017) with 3D reconstructions oriented and scaled with GPS ground control points and calibrated against Google Earth imagery. This yielded a final photorealistic 3D mesh (digital outcrop) comprised of 2.9 million mesh triangle faces, with an average ground pixel resolution of 0.24 m and total coverage of ~ 1.5 km2.

3. 244 ground-based digital images, of sub-mm (0.1-0.3 mm) ground-pixel resolution were collected at outcrop using a handheld DSLR camera during fieldwork. Handheld camera images used in this study for fracture orientation characterization were collected by Watkins et al. (2019) for fracture intensity analysis. Photographs of fractured bedding surfaces were acquired at a distance of 0.5 – 1.5 m from the outcrop, along a series of transects across the crest of the structure. Imagery was acquired along with GPS coordinates and camera orientation data at each photo location, allowing images to be georeferenced and re-oriented prior to manual digitisation of fracture traces.

Manual digitisation of fracture traces in 2D (satellite and ground-based images) and 3D (via digital outcrop) was performed in Move 2016.1 (formerly Midland Valley, now Petroleum Experts). Orientations of digitized fracture traces were extracted using FracPaQ 2.3 (Healy et al., 2017) and Move 2016.1. 2D fracture intensity was calculated from digitized fracture





traces by calculating total fracture length per unit area in 2D [m/m2]. Fracture intensity calculations were carried out in Move 2016.1 for handheld camera images and in ArcMap 10.5.1 (ESRI) for satellite and digital outcrop data.

## 5 Results

### 5.1 Lithostratigraphy


Approximately 78 meters of distinctly bedded, partially dolomitized bioclastic limestones of the Dupuyer Creek Unit (Castle Reef Formation) is exposed in the dam-cut at Swift anticline (Figs. 5A, B). The exposed interval has been subdivided into several informal lithological units based on carbonate lithology and facies. Units C1-C8 are exposed in cross-section view only and overlying unit S1-S5 are exposed across the crest of the structure (Fig. 5A). Compositional and textural variations

described by Watkins et al. (2019) in the exposed interval mainly reflect cyclical variations in depositional facies (e.g., Nichols, 1984). Bioclasts within coarse-grained units (grainstones and packstones) are dominated by dolomitized crinoid fragments. These grainstones and packstones are commonly structureless or marked by distinct planar cross-bedding (e.g., Unit C5; Fig. 5A). Fine-grained units (e.g., Units S2 & S3) are generally characterised by planar lamination, the presence of chert nodules, and microcrystalline textures. These mudstones or wackestones are generally mud-supported and commonly include large

(several cm), isolated colonial corals, particularly in the upper part of the exposed section (Fig. 5A). Coarser-grained packstones and grainstones at Swift anticline generally record greater bed thicknesses (1.4 m – 18 m) than fine-grained wackestone lithologies (0.7 m – 2.6 m).

### 5.2 Fold geometry and field observations

A cross-sectional view of the anticline at Swift Reservoir dam (Fig. 5B) provides an overview of the fold geometry: the

shallowly-dipping to horizontal backlimb transitions abruptly through a relatively narrow hinge zone to a steeply-dipping to vertical forelimb. Thrusts and back-thrusts, with relatively low offsets (> 0.5 m), are common through the exposed section (Fig. 5B). The cross-section view records a general trend for back-thrust dominance in the hinge and forelimb of the anticline, with thrusts better developed in the backlimb of the structure. The fold geometry and patterns of thrusting may vary significantly through the structure; the dam cut cross-section, however, provides the best available cross-sectional view of

Swift anticline.

3D fracture orientations collected at the dam cut cross-section during fieldwork, and subsequently from a high-resolution digital outcrop of the same locality, record a range of fracture orientations (Fig. 5C). Most easily identified and measured at the dam cut cross-section are the thrusts and back-thrusts which typically have a strike orientation parallel to that of bedding, with dips that range from sub-horizontal to ca. 50 degrees. An extensional fracture set, interpreted to be related to outer-arc

extension of folded strata, is well developed through the exposed cross-section. This fracture set is strike parallel to both bedding and the mapped thrusts and back-thrusts (Fig. 5C).





**Figure 5: (A)** Stratigraphic log through exposed units at the southern side of the Swift Reservoir dam cut. **(B)** Cross-sectional field image of dam cut showing general fold geometry, interpreted thrusts and back-thrusts, and approximate position of structural and stratigraphic log in A. Note, units S1 to S5 defined as those exposed on the crest of the structure (see Fig. 10). C1 to C8 units exposed only in cross-section view in **(B)**. Mm = Mississippian Madison Group; Je = Jurassic Ellis Group. **(C)** Orientation data from field and digital outcrop measurements showing NW-striking bedding planes, thrusts, back-thrusts and outer-arc extension fractures. Fracture classifications in **(C)** based on field observations and fracture orientations.



## 5.3 Fracture attributes from Google Earth imagery

Fracture mapping of satellite imagery (0.3 – 0.4 m ground pixel resolution) was carried out using images downloaded from Google Earth. 2717 linear features were identified as fractures and digitized from satellite imagery (Fig. 6A). A rose plot of 2D fracture orientations by trace count (number of mapped fractures) record an approximately bimodal directional distribution with two dominant fracture sets oriented approximately parallel (NNW-SSE) and perpendicular (ENE-WSW) to the fold axial trace (Fig. 6A). A similar trend was reported from field-based measurements by Watkins et al. (2019).

**Figure 6: (A) Manually interpreted fracture trace map from satellite imagery (© Google Earth/Landsat/Copernicus; pixel resolution = ca. 0.35 m). Rose plots show orientation distributions for all fractures mapped at this scale by fracture count (upper) and by cumulative length (lower). (B) Estimated 2D intensity (m/m2) of fractures mapped from satellite imagery. 2D fracture intensity calculated using the *Line Density* tool in *ArcMap 10.5.1* with 5 m grid cells and 50 m sampling window radii.**

The length-weighted rose plot (histogram of summed lengths) of the same fracture traces (Fig. 6A) shows the greater lengths of N-S oriented fractures. Although the N-S fractures do not appear to make up a significant component of the fracture population by count, length-weighting the data shows the importance of these features as a contributor to the overall population. Bulk fracture intensities (total fracture length per unit area for all mapped fractures) from satellite image data show a general





increase in fracture abundance towards the hinge zone of the anticline, particularly in central and southern domains (Fig. 6B). Increased fracture intensity values (e.g., > 0.4 m/m2) cluster along and around the fold axial trace forming discontinuous patches of high intensity fracture zones along strike that are not exactly coincident with the axial trace of the anticline.

Fractures mapped in satellite imagery were assigned to one of six discrete fracture sets (A-F) based on their orientations with respect to the orientation of the interpreted fold hinge line proximal to the interpreted fracture (see insets in Fig. 7A). Because the fold hinge exhibits some orientation variability along its length(Fig. 7A), some overlap exists between the orientations of the assigned fracture sets due to variability in the orientation of the fold hinge (Fig. 7B). Set A fractures are oriented ENE-WSW (mean strike = 59o), approximately perpendicular to the axial trace of the anticline, Set B fractures strike approximately parallel to the fold hinge (NNW-SSE; mean strike = 154o), and Sets C, D, E, and F are oriented approximately WNW-ESE, N-S, E-W, and NNE-SSW, with mean strikes of 111o, 178o, 086o, and 024o respectively (Fig. 7B). Fold-perpendicular (Set A) and fold-parallel (Set B) fractures make up the majority (ca. 40% and 31.5% respectively) of the total number of fractures mapped in satellite imagery. This dominance of Sets A and B accounts for the approximately bimodal orientation distribution for all combined fractures (Fig. 6A) and the overall trend for increased bulk fracture intensity towards the fold hinge. The remaining fracture sets make up 28.5% (by count) of mapped fractures from satellite imagery, with Sets C, D, E, and F representing 13%, 9%, 5.5%, and 1% of the total number of mapped fractures respectively.

Length distribution data show that with the exception of Set F, fractures in all sets exhibit relative increases in fracture abundance at shorter length scales (see downward-widening violin plots in Fig. 8). The predominance of relatively short fractures as a proportion of the total is most pronounced in Sets A and B, as evidenced by the width of violin plots at lower length scales. Set B has the lowest minimum (1.7 m) and median (14.7 m) fracture length values and Set D (oriented N-S) contains the longest fractures, with a maximum fracture length within this set of 515 m.

Estimated fracture intensities for the separated fracture sets provide an overview of how fracture abundance within each set varies spatially (Fig. 9). Set A fracture intensity data show some evidence for increased intensity towards the fold hinge, but this increase is neither uniform across the fold, nor are increased intensities exactly coincident with the interpreted fold hinge position (Fig. 9A). This suggests that proximity to fold hinge only partially controls the abundance of this hinge-perpendicular fracture set. Fractures of Set B (NNW-SSE) appear to be strongly developed along the hinge of the anticline, and increases in Set B intensity appear to be closely related to the position of the fold hinge (Fig. 9B). Backlimb positions exhibit low to moderate intensities of this hinge-parallel fracture set, with only isolated patches on the backlimb showing elevated intensity values (up to 0.07 m/m2). Sets C and D show evidence for isolated zones of increased fracture intensity but in both cases increased intensities do not appear to be systematically related to the fold hinge position (Fig. 9C, D). Fractures of Sets E and F were only identified and mapped in isolated parts of the structure. These sets show no systematic increase in abundance at hinge/forelimb positions and therefore changes in the abundance of these fractures are not easily related to fold geometry or structural position (Fig. 9E, F).





**Figure 7: (A)** Fracture traces separated into 6 discrete sets, based on orientations of individual fracture traces with respect to the fold hinge orientation. Rose plot in (A) shows fold hinge orientation variability along the crest of the structure, based on 15 line segments of the interpreted fold hinge. Insets in (A) show along-strike variability in fold hinge and hinge-perpendicular fracture orientations. Despite this variability in absolute orientations of hinge-perpendicular fractures, these were assigned to the same fracture set (Set A). **(B)** Equal area, length-weighted rose plots showing orientations of interpreted fracture sets. Orientation variability and overlap between fracture set orientations is attributed to fold hinge orientation variability (see insets in A).



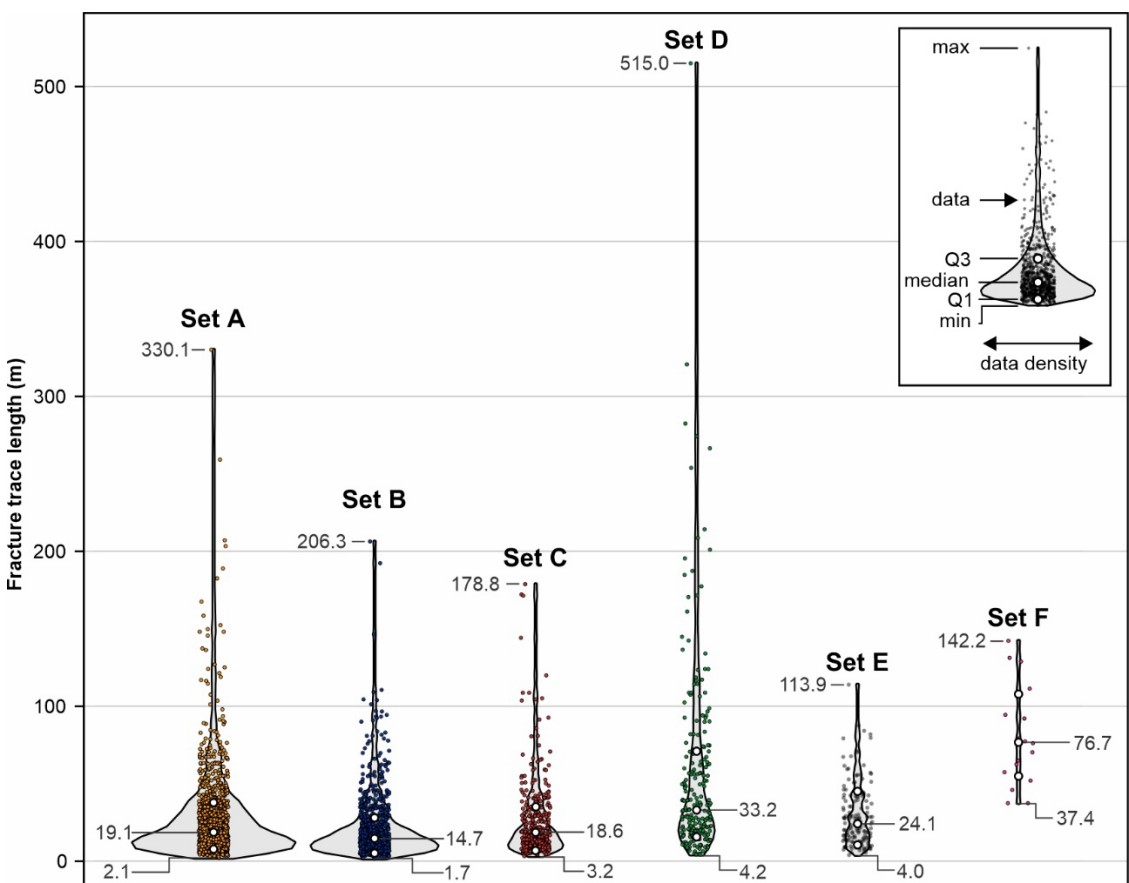

**Figure 8: Violin plots showing length distributions of fractures sets interpreted from satellite imagery. Numbers refer to maximum, median, and minimum lengths for each fracture set, in meters. Fracture sets and orientations shown in Figure 7.**

## 5.4 Digital outcrop analyses

Subsequent to initial mapping of fractures in satellite imagery, a second stage of fracture mapping was carried out using the digital outcrop of Swift anticline. Analysis of the UAV-imagery-derived photogrammetric reconstruction (digital outcrop) focused on (i) identifying of stratigraphic exposure levels and boundaries across the crest of the structure, (ii) remapping of fracture traces at a higher resolution in order to refine the fracture map and compare results with fractures mapped from Google Earth satellite imagery, and (iii) assessing of the relationship between structural position, mapped lithologies, and fracture attributes.

### 5.4.1 Digital outcrop derived lithology maps

The 3D digital outcrop allows lithological boundaries that are not clearly visible in satellite imagery (e.g., Fig. 4A) to be identified in 3D and mapped across the outcrop (see https://tinyurl.com/2hw54793 for low resolution, web version of the



photogrammetric reconstruction). Lithological boundary maps were generated by interrogating the digital outcrop in 3D and identifying lithological boundaries based on variations in texture, colour, and topography across the structure.

**Figure 9: (A-F) Estimated fracture intensity for fracture sets A-F. See Figure 7 for fracture set orientations. Fracture intensity maps generated using the *Line Density* tool in *ArcMap 10.5.1* with 5 m cell sizes and 50 m search radii. Approximate position of the anticline hinge shown by thick black and white line.**



Units S1-S5, defined in physically measured stratigraphic section in the upper part of the dam cut section (Fig. 5), are exposed in fractured bedding pavements on the crest of Swift anticline and were mapped digitally, using the methodology outlined above, across the exposed parts of the structure (Fig. 10A). Unit S4 makes up the majority of the exposure surface across the crest, particularly at backlimb structural positions (Fig. 10A). Units S1, S2, S3, and S5 are discontinuously exposed across the structure, and in some cases are only sufficiently exposed for fracture mapping in a single structural position (e.g.,

unit S1 in forelimb position, Fig. 10A). It should be noted that lithology mapping away from the measured section at the dam was undertaken using a digital approach only, with no ground-truth data collected to confirm digital lithology mapping results. As such, the lithology map in Figure 10 likely represents an oversimplification of the exposed bedding surface map. Patches of the mapped outcrop exposure may represent thinly bedded layers between our assigned units (S1-S5) but nevertheless, our detailed digital mapping and interrogation of the digital outcrop in 3D is interpreted to have resulted in a lithology map that

provides a good approximation of the lithologies exposed in the dam cut (Fig. 5) and on the crest of the structure.

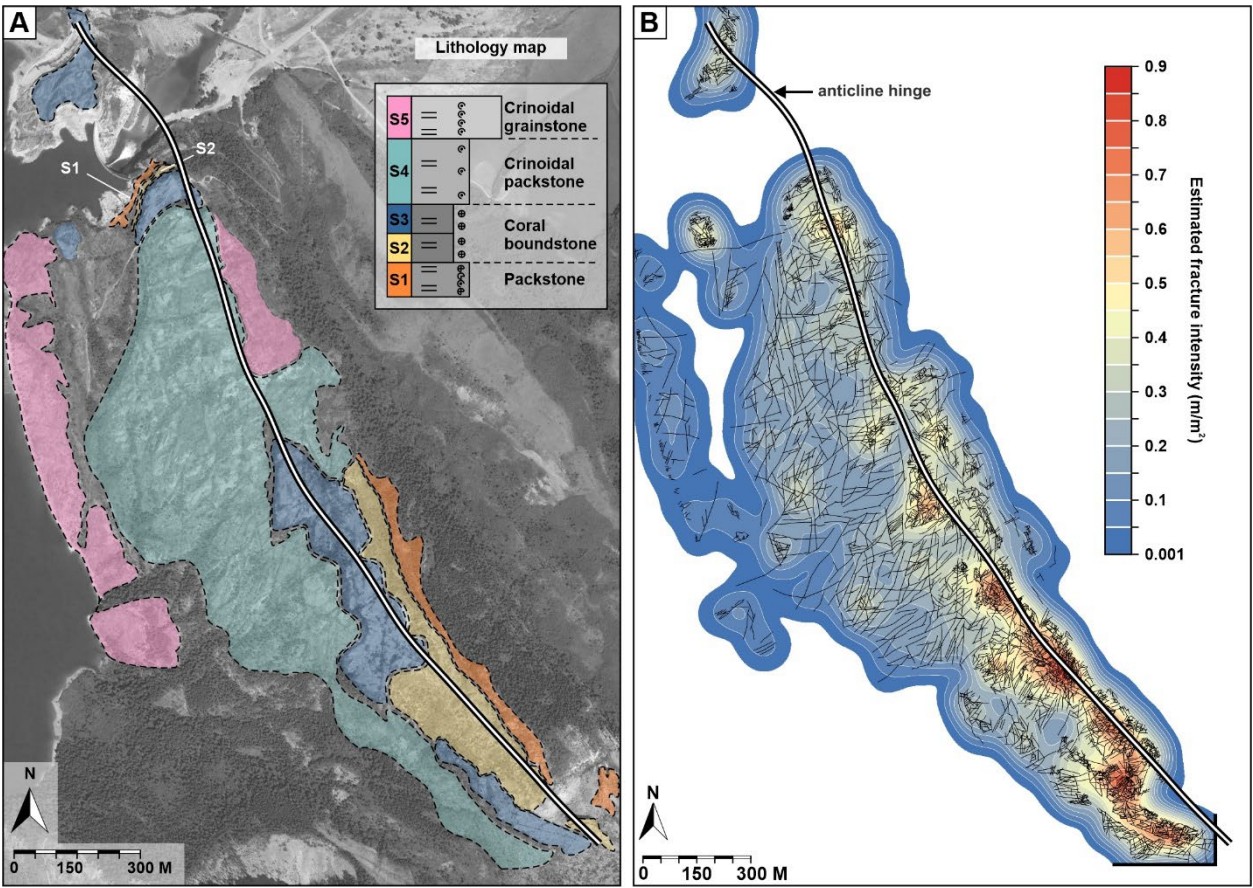

**Figure 10: (A) Lithology map of Swift Anticline (see Fig. 5A, B for stratigraphic log) overlain on satellite imagery (© Google Earth/Landsat/Copernicus). Lithological mapping was performed using the digital outcrop of the anticline and projected onto**

**satellite imagery for clarity. (B) Estimated 2D intensity (m/m2) of fractures mapped from digital outcrop (photogrammetry) data. 2D fracture intensity calculated using the *Line Density* tool in *ArcMap 10.5.1* with 5 m grid cells and 50 m sampling window radii.**



### 5.4.2 Estimated fracture intensities from digital outcrop mapping

Digital mapping of fractures on the digital outcrop was performed in 3D using a medium resolution digital outcrop that covered the entire outcrop exposure at 0.24 m ground pixel resolution (compared to 0.3-0.4 m for Google Earth imagery). This second stage of fracture mapping resulted in identification and mapping of 4608 fractures (Fig. 10B), compared to 2717 fractures mapped in satellite imagery. Estimated fracture intensities for digital-outcrop-derived data are higher (up to c. 0.86 m/m2; Fig. 10B) than for equivalent Google Earth-derived data (up to c. 0.54 m/m2; Fig. 6B) but general trends in fracture intensity for the two datasets are similar. Both intensity maps (Figs. 6B and 10B) exhibit discontinuous patches of relatively high fracture intensity around the fold hinge line, but with variations in both strike and dip directions. Neither of the fracture intensity maps show a perfect match between the position of the interpreted fold hinge position and highest fracture intensities; in both cases the highest fracture intensities appear to be proximal to the interpreted hinge position, but a short distance (20-50 m) towards the backlimb of the structure.

A compiled lithology and fracture intensity map for the digital-outcrop-derived data (Fig. 11A) shows that variations in fracture intensity at Swift anticline are at least partially related to stratigraphic exposure level. There is a general trend for increased fracture intensity in the finer-grained, mud-supported units S2 and S3, as documented by fracture intensity values of 0.4 to >0.8 m/m2 where these units are exposed. Patterns of fracture intensity contours appear to closely correspond to the mapped extents of units S2 and S3, with highest fracture intensities present towards the geographic centres of these exposed units (Fig. 11A). It should be noted that there is likely an edge effect in calculated fracture intensity towards the edges of the exposure (Fig. 11) but nevertheless, units S2 and S3 exhibit the highest fracture intensities on the crest of the structure. Fracture intensity values generally decrease from unit S3 to units S4 and S5 irrespective of structural position. This is apparent where these units are exposed on the backlimb of the structure: S3, S4, and S5 are associated with fracture intensities greater than 0.4 m/m2, greater than 0.3 m/m2, and less than 0.2 m/m2 respectively (Fig. 11A). Unit S4 is the only unit that is well exposed at a number of structural (backlimb, hinge, and forelimb) positions. Fracture intensities in this unit increase towards the hinge of the structure (Fig. 11A), suggesting that structural position influences fracture intensity. In other units only isolated parts of the outcrop allow for comparison of fracture intensities in similar structural positions. The relative importance of structural position vs. lithological variations on fracture intensity is therefore difficult to assess but nevertheless, it appears from the data provided in Figures 10 and 11 that both of these factors play a role in observed fracture intensities at Swift anticline.



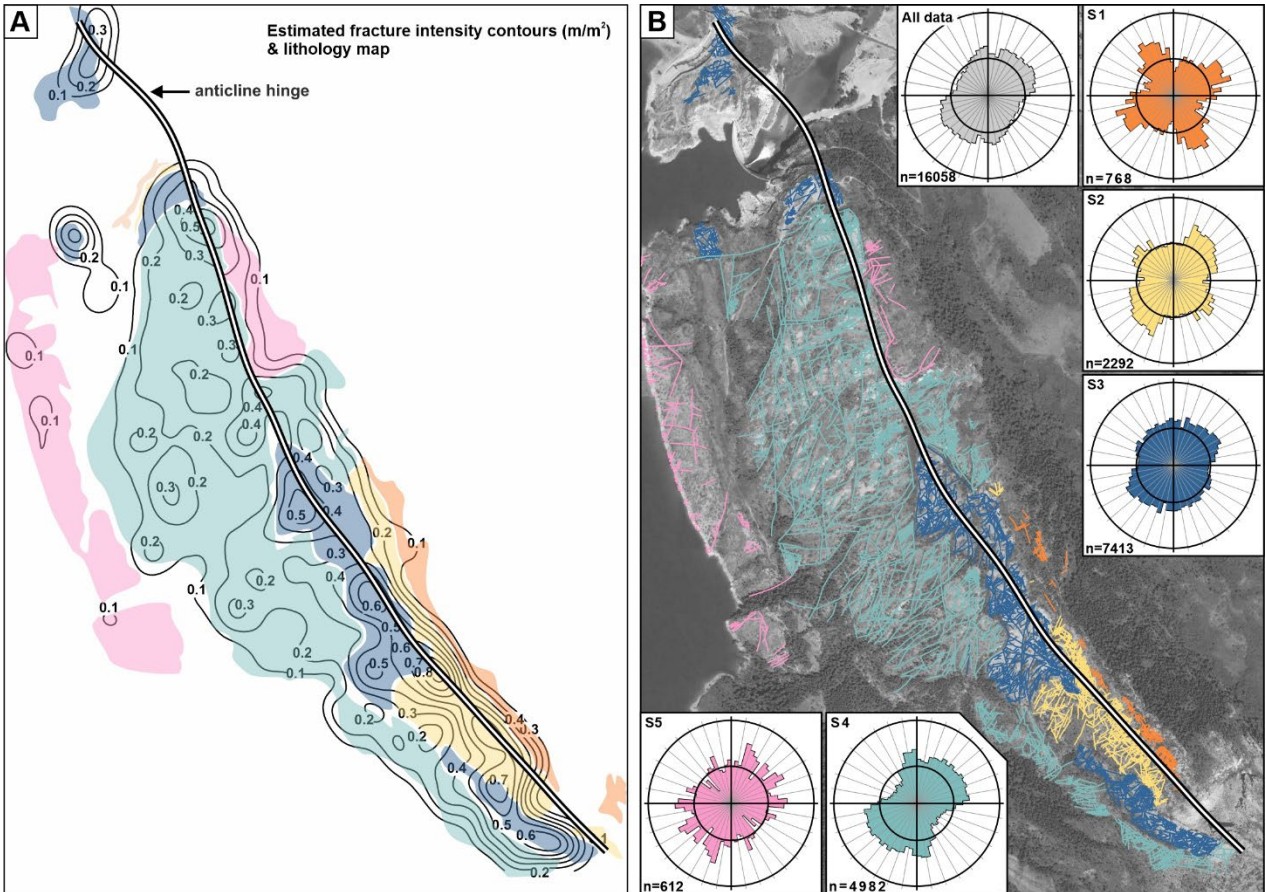

**Figure 11: (A) Digital-outcrop-derived lithology map overlain onto estimated fracture intensity contours derived from the digital outcrop. (B) Digital-outcrop-derived fracture traces (n = 4608) coloured according to unit in which they were mapped, overlain on satellite imagery (© Google Earth/Landsat/Copernicus). Orientations of fracture traces by lithology shown in rose diagrams. Note, n values shown for rose plots are for fracture trace segments (i.e., straight segments between polyline nodes), rather than for entire fracture traces. Satellite imagery © Google Earth/Landsat/Copernicus.**

### 5.4.3 Stratigraphic exposure vs. fracture orientations

Length weighted orientations for digital outcrop-derived fracture traces record a weakly preferred orientation of the fracture population of NE-SW (Fig. 11B), perpendicular to the fold hinge. This dataset records greater overall dispersion of fracture trace orientations than data derived from satellite imagery (Fig. 6A) which is indicative of greater variability in fracture orientations at smaller scales. Separation of fracture orientation data into stratigraphic units records changes in dominant fracture orientations with exposure level (Fig. 11B). Fractures traces within units S1 and S2 show dominantly bimodal distributions, with well-defined peaks in length-weighted fracture orientations that trend roughly NE-SW and NW-SE. Orientations within units S3-S5 record greater variability, with no clearly defined, dominant orientations. The approximately bimodal distributions recorded within units S1 and S2 (Fig. 11B) show the apparent dominance of hinge parallel and hinge





perpendicular fractures in these lithologies. Units S1 and S2 are exposed only in the hinge and forelimb zones of the structure,
in which overall fracture patterns from aerial imagery show an increasing abundance of fractures oriented perpendicular and
parallel to the axial trace of the fold (Fig. 9A, B). It is therefore likely that the observed changes in fracture orientations by
stratigraphic exposure level (Fig. 11B) reflect structural rather than stratigraphic controls.

### 5.5 Fracture orientations from high resolution (handheld camera) imagery

Watkins et al. (2019) assessed the influence of stratigraphic and structural factors on fracture intensity at Swift anticline.
The same fracture stations as used by Watkins et al. (2019) are used here to assess fracture orientation variability in high
resolution (0.2 mm pixel size) imagery.

Stacked fracture orientation histograms along a series of structural transects, derived from field-based orientation
sampling, provides an overview of spatial variability in dominant fracture orientations across the structure (Fig. 12). Field data
show a general trend for increased dominance of hinge-parallel fractures towards forelimb and hinge positions on the anticline.
This trend is clearer in the central part of the structure (e.g., transects 3-7) where orientation histograms display distinct peaks
towards SE and SSE, approximately parallel to the fold hinge axis (Fig. 12). This trend is not ubiquitous however; fracture
orientations at some hinge and forelimb positions, particularly towards the southern part of the structure (e.g., transects 8, 9,
10), show no clear dominance of hinge-parallel fractures, and relatively dispersed orientations.

Strike perpendicular fractures (set B, Fig. 7B) are less prominent in field data but do show dominance in some isolated
positions (e.g., backlimb, transect 1, Fig. 12). Fracture sets C, D, E and F, identified from satellite imagery (Fig. 7), are not
clearly evident in field-based orientation histograms but do make up a component of dispersed fracture orientations,
particularly towards the southern part of the structure. It should be noted that transects 7, 8 and 9 do not sample the forelimb
of the structure, and thus the increasing dominance of hinge-parallel fractures, mapped in satellite imagery (Fig. 9), towards
the forelimb of the structure may not be represented here due to a lack of exposure. Many of the ground stations, when observed
in isolation, show no clearly dominant fracture orientation; general trends are only apparent when multiple histograms are
stacked along structural transects.

### 5.6 Scaling of fracture orientations

To assess the impact of observation scale on apparent fracture orientations, data from ground sampling sites were
compared with digital outcrop-derived (within a 50 m sampling circle) and satellite image-derived (100 m sampling window)
data around field measurement stations (Fig. 13). This window sampling method allows for comparison of fracture orientations
between field-based, digital outcrop, and satellite image observation scales. Fractures described in the section below are
referred to according to their orientations and assigned fracture sets, as outlined in Figure 7.







**Figure 12: Stacked orientation histograms, from field-based measurements, along a series of structural transects across the anticline. Interpreted structural positions are marked along upper horizontal axes of the transects. Yellow and red bars show approximate fold-perpendicular and fold-parallel orientations respectively for structural transects. Approximate orientations of yellow and red bars at each transect are derived from the interpreted fold axis in Figure 6B. Black dots and boxes on map show sample locations and data used for each transect, respectively. Satellite imagery © Google Earth/Landsat/Copernicus.**







Field-derived orientation data generally record the dominance of hinge-parallel (set B) fractures in the forelimb and hinge of the structure (Fig. 13). Hinge-perpendicular (set A) fractures are also commonly sampled at these locations but are generally

less well developed than set B. Slight variability in the orientation of sets A and B between sample sites at forelimb and hinge positions is attributed to variation in the orientation of the fold axial trace (e.g., sample sites 1, 2, and 3). Field data from the backlimb of the structure show greater variability than in hinge-proximal positions, with no consistently dominant fracture orientations observed. In general, individual sample sites on the backlimb show greater apparent orientation dispersion (e.g., sites 11, 12, and 13) than forelimb and hinge counterparts. Marked variations in fracture orientations are also recorded at

sample sites that are adjacent to each other (e.g., sites 10 and 16) towards the fold backlimb.

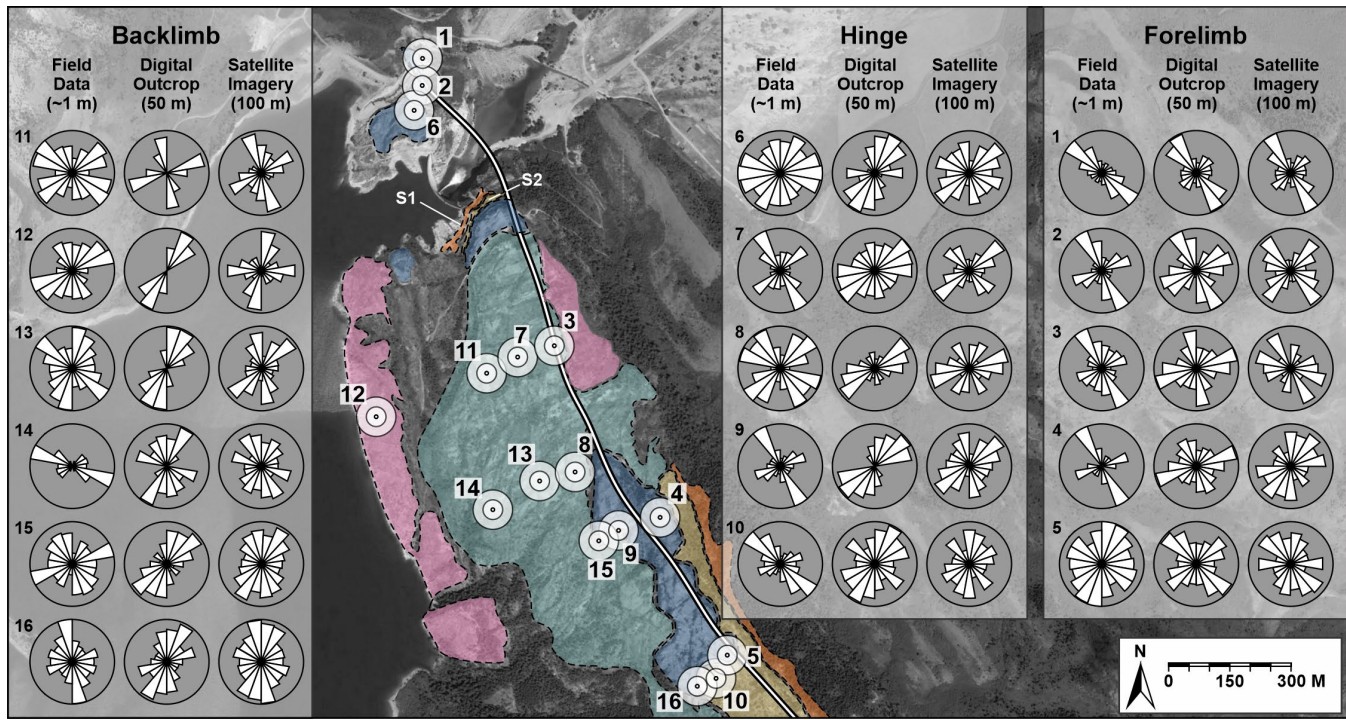

**Figure 13: Comparison of fracture orientations at multiple observation scales. Bullseye circles on satellite image represent**
**approximate sampling areas for field, digital outcrop, and satellite data from smallest to largest respectively. Rose plots show variation in average fracture orientation with changes in observation scale at each site. Numbers on the satellite image correspond to rows of rose plots in shaded boxes. See Figure 10 for lithology colours. Satellite imagery © Google Earth/Landsat/Copernicus.**

Window samples of digital outcrop and satellite image derived fracture traces show some similarity to field data at the

fold forelimb (e.g., site 1; Fig. 13). Orientation distributions are generally dominated by sets A and B in forelimb digital outcrop and satellite image sampling windows, as is the case for field data (e.g., sites 1 and 2; Fig. 13). In some cases sets A and B are both recorded, but there exists a difference in fracture set dominance with observation scale. At site 2, for example, field data



show the dominance of a NNW-SSE component, while digital outcrop and satellite image derived fractures within sample windows around the site demonstrate an approximately bimodal orientation distribution (Fig. 13). In general, sample window
data from the fold hinge and backlimb show less agreement to field data than in the forelimb and hinge of the structure. In some cases, one or more of the fracture sets is represented at all three observation scales (e.g., site 7, set A; Fig. 13), while in others, little similarity exists between datasets extracted from the same area (e.g., site 11; Fig. 13). In general, there appears to be greater agreement between observation scales where sets A and B are more strongly developed, primarily in hinge-proximal locations. General trends in fracture orientations, either from field-based sampling or by window sampling of remotely
acquired data, are difficult to identify from isolated sampling of the structure due to variability in the fracture network at a range of scales.

**5.7 Scaling of estimated fracture intensity**

Watkins et al. (2019) employed a field-based approach to characterize fracture intensity at Swift anticline by collecting handheld imagery of fractured bedding surfaces and using the circular scanline method of Mauldon et al. (2001) to estimate
fracture intensity in 193 scaled and oriented field images. The authors found that fracture intensity varies substantially at Swift anticline, and that both lithology and structural position influence fracture occurrence. Here we compare our results with those of Watkins et al. (2019) by sampling our satellite image and digital outcrop derived fracture intensity maps (Figs. 6B and 10B) at the precise sample site locations used for the previous study. Fracture intensity map sampling was performed using raster sampling tools (Extract Values to Points) in ArcMap 10.5.1.
Estimated fracture intensity varies substantially according to image pixel size and scale of observation (Fig. 14), with fracture intensity estimates of ca. 24 to 463 m/m2 for handheld camera images (pixel size = ca. 0.2 mm), 0.0026 to 0.69 m/m2 for digital outcrop data (pixel size = ca. 0.24 m), and 0.0003 to 0.33 m/m2 for satellite image data (pixel size = ca. 0.35 m). Power-law regression fits for minimum, median, and maximum fracture intensity values for compiled data provide regression coefficients of 0.98 and higher, and power law exponents of 0.9 to 1.4 (Fig. 14). Although we only assess only three image
pixel sizes in this analysis, the high correlation coefficients for power-law regression models suggest that prediction of fracture intensity for a given observation scale may be relatively well constrained.



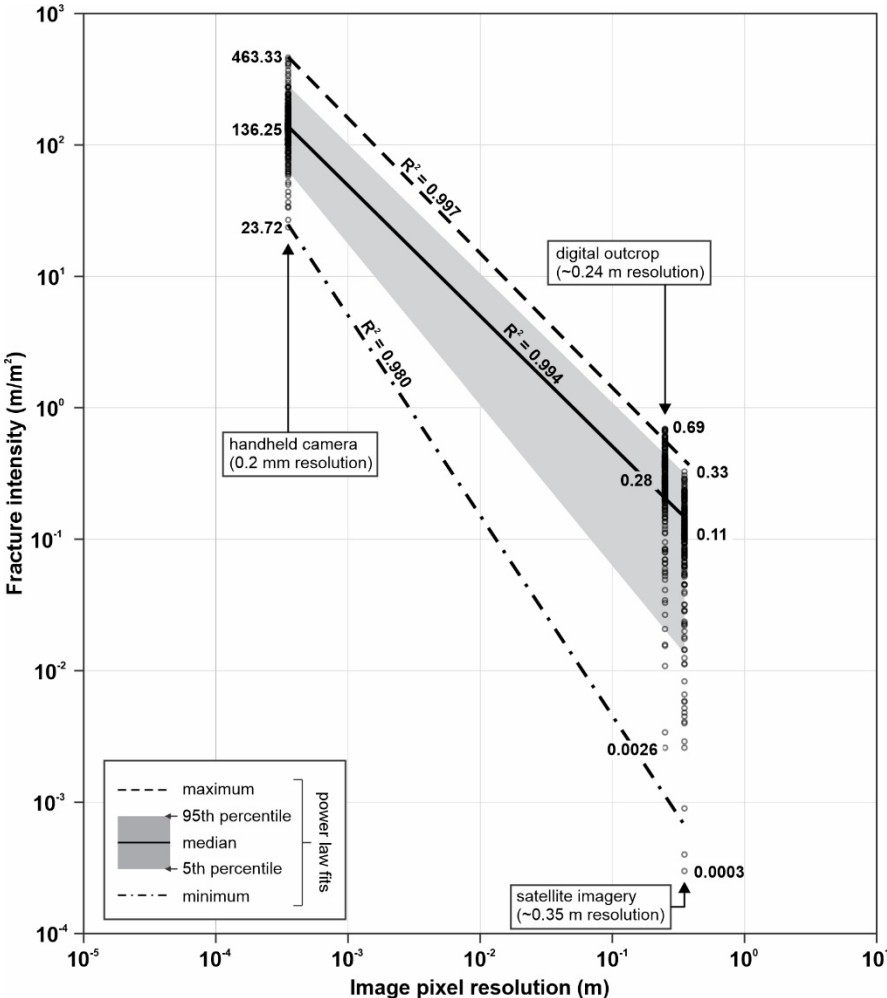

**Figure 14: Compilation of estimated fracture intensities and ground pixel resolutions for Swift Anticline. Fracture intensities estimated from handheld camera images are reproduced from Watkins et al. (2019).Satellite imagery and digital outcrop data generated by sampling fracture intensity rasters (Figs. 6B and 10B) using the geographic coordinates of Watkins et al. (2019) field localities. Labels denote the maximum, median, and minimum fracture intensity values for the respective datasets. Raster sampling was performed in ArcMap 10.5.1.**

# 6 Discussion

## 6.1 Controls on estimated fracture intensity

Our results show that apparent natural fracture intensity at Swift Anticline is controlled by both stratigraphic exposure level and structural position. Digital fracture maps and associated intensity contours show that fine-grained, mud-supported wackestones are the most intensely fractured lithologies at the site (Fig. 11). This result generally agrees with the findings of Watkins et al. (2019) who report increased fracture intensities in mud-supported units at the site. These authors suggest that higher fracture intensities in mud-supported lithologies may be due to relatively low porosities (and associated increases in



rock strength) in these units. Previous studies have shown that rock strength generally decreases with increasing porosity (e.g., Price, 1966; Dunn et al., 1973; Nelson, 2001) and that fine-grained, low porosity lithologies may be more brittle, and therefore more prone to intense fracturing than coarse-grained rocks (e.g., Hugman and Friedman, 1979; Wennberg et al., 2006; Hanks et al., 2007). Our observations of increased fracture intensity in fine-grained units are in agreement with the work of Watkins et al. (2019) and others, and provides a relatively simple but logical link between rock texture and fracture intensity. Other

lithological properties (e.g., bed thickness) may influence fracture abundance (e.g., McQuillan 1974; Ladeira and Price, 1981; Wennberg et al. 2006; Sun et al., 2021) but these factors are not assessed here.

A limitation of the analysis of Watkins et al. (2019) is that specific lithologies (e.g., mud-supported units) identified in the field could not be easily correlated across the exposed crest of the structure. By digitally mapping lithological boundaries across the structure in 3D (Fig. 10A), spatial variations in fracture properties can be directly tied to stratigraphic exposure

levels and therefore larger-scale, three-dimensional assessments of fracture intensity vs. stratigraphic exposure can be more easily performed using the digital approach (e.g., Corradetti et al., 2018; Triantafyllou et al., 2019). It should be noted that limits to image resolution may partly hamper lithologic boundary mapping (e.g., Humair et al., 2015) and while every effort was made to generate robust interpretations in this study, we acknowledge that delineating precise boundaries between units was not always straightforward. A potential solution to this problem, resources allowing, would be to carry out initial digital

mapping or reconnaissance of sites using satellite imagery or photogrammetric reconstructions, followed by field campaigns focused on further data collection and field-checking of digital interpretations (e.g., Scheiber et al., 2015).

We provide evidence for increased fracture abundances towards the hinge zone of the anticline (Figs. 6 and 10). These results are generally consistent with the results of Watkins et al. (2019) but in both this and the previous study the relationship between structural position and fracture intensity is not straightforward (e.g., Fig. 6 in Watkins et al., 2019). We show that

fracture intensity values generally increase towards the fold hinge but that the zones of highest fracture intensities are not always perfectly coincident with the interpreted hinge position (Fig. 6B and 10B). Increased fracture abundances in hinge-proximal zones have been recorded on folds in a multitude of settings (e.g., Ramsay, 1967; Hanks et al., 1997; Hennings et al., 2000; Wennberg et al., 2007; Ghosh and Mitra, 2009; Watkins et al. 2015) but, as noted above, lithology also influences estimated fracture intensity at Swift Anticline. Because multiple stratigraphic units are exposed on the crest of the structure,

our fracture intensity maps (Figs. 6B and 10B) record the influence of both structural position and stratigraphic exposure level on fracture abundance. Where multiple lithologies are exposed on fold structures, it should perhaps be expected that apparent fracture intensity does not directly correlate with either structural position (e.g., forelimb vs. backlimb) or proximity to the fold hinge.

Our data show that estimated fracture intensity generally increases at higher image resolutions (Fig. 14). Previous studies

have found similar trends in fracture intensity data derived from remotely acquired data, with relationships generally conforming to power law distributions (e.g., Castaing et al., 1996; Odling, 1997; Bonnet et al., 2001; Bossennec et al., 2021; Chabani et al., 2021; Ceccato et al., 2022). Our somewhat limited dataset (with only three observation scales) has power law exponents of 0.9 to 1.4 for fracture intensity vs. pixel size (Fig. 14), which is within the range of exponents reported for



equivalent data in previous studies (e.g., Odling, 1997; Scheiber et al., 2015; Ceccato et al., 2022). Assessing fracture intensity
vs. image resolution was not the primary focus of this study and a more thorough assessment of resolution effects on fracture intensity would require intermediate resolution scales for robust determination of power law exponents. It is likely that the relationship between apparent fracture intensity and pixel resolution varies substantially according to a range of factors including rock type (e.g., siliciclastic vs. carbonate), fracture failure mode (e.g., shear vs. opening mode), fracture fill (e.g., quartz vs. calcite), fracture aperture, and degree of weathering, among others. Future studies focused on isolating the role of
these different factors would be valuable additions to the remote-sensing-derived fracture analysis literature.

## 6.2 Fracture orientation variability

Natural fracture orientations at Swift Anticline are highly variable and appear to vary according to stratigraphic exposure level (Fig. 11), structural position (Figs. 9 and 12), and observation scale (Fig. 13). Established conceptual models of discrete fracture sets on contractional folds (e.g., Price, 1966; Stearns, 1969) do not capture or predict the variable and dispersed fracture
orientations observed at Swift Anticline and our results therefore question the predictive potential of these models. We do, however, document at least two systematic fracture sets that appear to conform to established models of fold-related fracturing. Most of the Set A (fold-axis-perpendicular) and Set B (fold-axis-parallel) fractures observed on the crest of the anticline (Figs. 7 and 8) are interpreted as being fold related. These sets exhibit (i) a general increase in intensity towards the interpreted fold hinge position, and (ii) orientations that are consistently parallel (Set B) and perpendicular (Set A) to the local fold hinge
orientation (Fig. 7A). Opening-mode fractures oriented parallel and perpendicular to fold hinges have been documented on contractional anticlines in a number of settings (e.g., McQuillan, 1974; Bergbauer & Pollard, 2004; Cooper et al., 2006; Wennberg et al., 2006; Francioni et al., 2019), including at sites proximal to Swift Anticline in the Sawtooth Range (e.g., Stearns, 1964; Ghosh & Mitra, 2009).

The hinge-parallel (Set B) fractures at Swift Anticline are generally interpreted to have formed as opening mode fractures
in response to outer arc bending of relatively competent carbonate strata during fold formation, consistent with predicted bending strain on contractional folds (e.g., Ramsay, 1967). While most fractures observed at the site show no evidence for displacement parallel to the fracture walls (i.e., shear), the cross-section exposure of the anticline (Fig. 5B) exposes several thrusts and back-thrusts with similar strike orientations to fracture set B (Figs. 5C and 7). It is therefore possible that a small proportion of the Set B fractures observed in map view (Fig. 7) are reverse faults rather than bending-related opening-mode
fractures. The hinge-perpendicular (Set A) fractures observed in the field typically exhibit opening mode kinematics and these may have developed as a result of extension parallel to the fold hinge. Subtle along-strike plunge variations of the fold and associated hinge-parallel curvature (e.g., Cosgrove and Ameen, 2000) may be a potential mechanism for this hinge-parallel extension. We did not find any clear relationship between hinge-parallel curvature and hinge-perpendicular fracture intensity at the site, however, because of difficulties in accurately estimating fold curvature from the vegetated, eroded fold crest. The
underlying mechanisms that led to the development of Set A fractures is therefore somewhat speculative.



Fracture Sets C, D, E, and F exhibit no clear relationship between fracture intensity and proximity to the fold hinge (Fig. 9) and are less consistently oriented with respect to the fold hinge compared to Sets A and B (Fig. 7). From these general patterns, we tentatively interpret Sets C, D, E, and F as having developed prior to, or possibly after, fold formation. The units exposed on the crest of Swift Anticline were deposited during the Mississippian (Nichols, 1984), and based on tectonic

frameworks for North America in many published sources (e.g., Marshak et al., 2000; Weil and Yonkee, 2023), these rocks likely experienced variable regional stress through time related to multiple Late Paleozoic through Paleogene convergent tectonic events in western North America, although only the late Cretaceous to Palaeocene development of the Sawtooth Range (Fuentes et al., 2012) resulted in any significant contractional shortening. The strata exposed at the site have also experienced documented localized forebulge-related extension during the Middle Jurassic (Ward and Sears, 2007), and probably regional

extension (eastern Basin and Range province) during the Cenozoic (e.g., Wallace et al., 1990; Stewart et al., 1998). The units exposed at Swift Anticline have therefore experienced at least two phases of prolonged regional contraction, one phase of localized extension, and one phase of regional, plate-scale extension. Because of this complex tectonic evolution and the likely variations in principal stress orientations, magnitudes and fracture failure modes, and fracture orientations that this deformation history implies (Ferrill et al., 2021), it should perhaps be expected that multiple fracture sets could have developed both before

and after fold formation at the site. The existence of pre-folding fractures could have resulted in the reactivation of optimally oriented fracture sets to accommodate strain during folding, that may not directly conform to the expected orientation in conceptual models. A further consideration is the curvilinear nature of the fold hinge line and the implications for strain and fracture set development, as compared to the models derived for linear folds.

**6.3 Predicting fracture intensities and orientations**

Fracture orientations derived from field images, digital outcrop data, and satellite imagery show a general trend for increased proportions of hinge-parallel and hinge-perpendicular fractures towards the anticline hinge and forelimb. In contrast, more dispersed and less predictable orientations are present towards the backlimb of the structure (Fig. 13). This overall pattern results in a more clearly defined structural grain in the hinge and forelimb, and as such, greater agreement between observation scales at these structural positions. Less systematic or dispersed fracture orientations on the backlimb results in greater disparity

in orientations between observation scales, and a general trend for disagreement between data derived from field images, digital outcrop data, and satellite imagery (Fig. 15). These results suggest that extrapolation or prediction of fracture orientations from one observation scale to another is not straightforward, and that the scaling of fracture properties is dependent on both structural position and deformation history, among other factors.

Our observations of a more clearly defined structural grain in hinge and forelimb positions are similar to those of Watkins

et al. (2015, 2018), who showed that in strata that have experienced a long and complex deformation history, fracture orientations are more consistent and predictable at hinge and forelimb positions, and generally unpredictable on backlimbs. Our conceptual model of fold-related fracturing acknowledges that complex fracture patterns are likely to exist in rocks that have experienced complex deformation histories, and that fold-related fracturing is more likely to overprint pre-existing



deformation fabrics in hinge and forelimb positions (Fig. 15). Further, we account for stratigraphic exposure level, based on

our observation that fine-grained, thinly bedded units at the site exhibit higher fracture intensity than coarse-grained, thickly

bedded units (Fig. 15). This model develops early conceptual models that predict highly organized, discrete fracture sets with

little orientation variability by (i) accounting for the effects of lithological properties on fracture abundance, (ii) acknowledging

that brittle deformation fabrics may form both before and after folding, and (iii) documenting that the scaling of fracture

properties is likely dependent on structural position.


**Figure 15: Conceptual diagram showing variations in fracture attributes at Swift Anticline. Results from this study provide evidence for three general trends in fracture attributes: (i) increasing intensity of hinge-parallel and hinge-perpendicular fractures towards the fold hinge which results in a bulk increase in fracture intensity in hinge-proximal positions, (ii) higher fracture**

**intensities in fine-grained, thinly-bedded units, (iii) variable fracture orientations in backlimb positions, with little agreement between observation scales, (iv) moderate to high intensity of hinge-parallel and hinge-perpendicular fractures and some agreement between fracture orientation measurements at hinge positions, and (v) moderate to high intensity of hinge-parallel and hinge-perpendicular fractures and good agreement between fracture orientation measurements in high, medium, and low resolution data towards the forelimb. Rose diagrams are a subset of data provided in Fig. 13.**




Our results and conceptual model suggest that accurate prediction of fracture properties requires analysis not only of lithologic and structural properties, but also of fracture property scaling, and the spatial variability of scaling relationships. Finally, predictions should account for all known deformation events and the effects that these events may have on existing fracture patterns present-day.

**7 Conclusions**

In this study we assess the effects of structural position, lithology, and variable data resolution on estimates of natural fracture network properties. By characterizing fracture intensities and orientations derived from mapping fractures at three image resolutions, we assess how interacting geological factors influence fracture development and the scaling of natural fracture systems. Our findings are relevant for estimating and extrapolating fracture properties in the subsurface, where data resolution and coverage are limited. The key findings of this study are:

1. Structural position influences fracture intensity at Swift anticline. Combined fracture intensity (for all orientations) generally increases towards the fold hinge, though this pattern is patchy, and the zones of highest intensity are not exactly coincident with the position of the fold hinge. Separation of fracture traces into discrete sets shows that only the hinge-parallel and hinge-perpendicular fractures exhibit systematic increases in abundance towards the fold hinge. In contrast, fractures not oriented parallel or perpendicular to the fold hinge show no systematic variations in abundance across the structure.

2. We document a general trend for increased fracture intensity in relatively fine-grained, thinly bedded units at the site. Variations in stratigraphic exposure level across the crest of the structure result in fracture intensity maps that capture both lithologic and structural elements. Mismatches between zones of highest fracture intensity and the fold hinge position are attributed to stratigraphic exposure level.

3. Fracture orientations at the site are highly variable, and only hinge-parallel and hinge-perpendicular fractures are consistently oriented with respect to the orientation of the fold hinge. Further, these fracture sets are consistently identified at all observational scales in the forelimb and hinge; and show increased intensity in these regions. Other fracture sets show less consistency between observational scales and no intensity relationship with fold position.

4. Fracture orientation data exhibit greatest agreement between observation scales at hinge and forelimb positions where the hinge-parallel and hinge-perpendicular fracture sets are best developed. Based on these results, we suggest that the scaling of fracture properties is likely to be dependent on structural position. Extrapolation of fracture properties from one scale to another should therefore account for variations in deformation intensity.

5. The application of simple, classical models of fracture development related to folding have been widely used by many previous studies but may not be appropriate where a proportion of the observed fracture population formed before or after folding. Rather than predicting discrete, highly organized fracture sets, we show that fold-related fractures impart orientation anisotropy to otherwise highly dispersed fracture orientations. This suggests that fracture prediction should account for all known deformation episodes and the effects these events may have on present-day fracture patterns.



## Author contributions

All authors performed data collection during the field work. AC collected UAV imagery and processed photogrammetry data.
Fracture mapping and data analysis was performed by AC. The draft manuscript and figures were prepared by AC and finalized after input and edits were provided by co-authors. Manuscript conceptualization by AC with input from co-authors through numerous discussions.

## Competing interests

The authors declare that they have no conflicts of interest.

## 620 Code/Data availability

Data are provided in the manuscript and will be uploaded as supplemental information in the final version.

## Acknowledgements

Petroleum Experts (formerly Midland Valley Exploration) is acknowledged for academic use of Move 2016.1 software.
Thanks to David Ferrill, Kevin Smart, Casey Nixon, and Paul Gillespie for helpful discussions regarding fracture patterns.
This study was carried out as part of a University of Aberdeen PhD supported by the NERC Centre for Doctoral Training in Oil & Gas (grant No. NE/M00578X/1 awarded to Cawood). Additional funding for fieldwork was provided by The University of Aberdeen Fold-Thrust Research Group.

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
