# Peer review of "Natural fracture patterns at Swift anticline, NW Montana: the influence of structural position and lithology from multiple observation scales"

_EGUsphere, 2023_

## Author Comment (AC1)

**REVIEWER 1 (ANONYMOUS): COMMENTS AND RESPONSES**

Reviewer comments in black, responses in blue. Where line numbers are specified in responses, these refer to the "track changes" version of the revised manuscript.

**GENERAL COMMENTS**

The paper describes a study of fracture patterns across a range of observation scales in folded rocks. The topic is of general interest, despite decades of work on fold-related fractures, and is within the scope of the journal. The paper is clearly written and well-illustrated. This study builds on earlier work in the area (Watkins et al. 2019).

**Accept. Thank you for the detailed comments and insightful feedback.**

There are points that could use some clarification, more nuance, or additional referencing.

**Accept. Specific comments and questions are addressed in detail below.**

The use of the 'scaling' terminology needs some clarification. Does this refer to the different scales of image acquisition and image resolution and their effects on how pattern look? Or is it related to some intrinsic size-scaling of the fracture patterns? It seems like the authors mean the former (figure 1). But the tile ('scaling of natural fracture patterns') seems to imply the latter (as in Davy et al. 2010, A likely universal model of fracture scaling and its consequence for crustal hydromechanics). There don't seem to be any classic looking 'scaling' plots or extrapolation from one scale to another for reservoir characteriztion (as in Hooker et al., 2009, Aperture-size scaling variations in a low-strain opening-mode fracture set, Cozzette Sandstone, Colorado, Journal of Structural Geology), unless this is what is meant by figure 14. Some clarification is in order.

**Accept. We have revised both the Introduction and Discussion to address this general comment. Detailed responses to individual comments are provided below.**

The Arc map based fracture density mapping should be of interest to many readers.

**Thanks, we hope readers find the ArcMap approach of use.**

The Introduction is clear and manages to cover the huge literature on the topic of fold-related fractures and fracture documentation pretty efficiently. But there are two related points that could be expanded. Under the impediments to fracture pattern analysis (45) the non-uniqueness problem of fracture analysis should be mentioned. In practical terms this is a central concern in using outcrop fractures for any purpose. Fracture attributes such as overall shape and even patterns that are independent of loading path or are not uniquely determined by the mechanical processes that formed them, obstruct clear interpretations of the structural history. Path-independent structures, as might be seen in individual fractures, either in outcrop or in core specimens, commonly cannot be ascribed to a unique mechanistic interpretation without additional information (like direct fracture dating). A path independent or nonuniqueness problem underlines some of the issues that limit fracture core analysis and is the main impediment to using outcrop fracture patterns reliably. Outcrop studies show that patterns can vary markedly with rock type and fracture formation

mechanism, but the path independence that makes outcrop fracture patterns hard to interpret also makes these patterns impossible to infer from a sample of an individual fracture in either a core or in an outcrop. This is a case of fractures, individually, being too simple (are they fold related or due to uplift and unloading, for example). An issue related to this is, how to guard against well characterized but misleading outcrop fracture patterns? Both issues are addressed at length in a recent review (noted below) and so maybe these points can be mentioned mostly via reference to the literature. But the ambiguity of outcrop observations needs to be addressed in the Discussion as well.

**Accept. We have modified the Introduction accordingly (Lines 66-76). Thank you.**

Also with respect to the Introduction, in my opinion the last paragraph (96-102) would be more compelling if it were to be restated as specific claims along the lines of 'here we show that…' Get the reader's attention by teasing the actual findings here. With so many papers on fold-related fractures, drawing the reader in seems like it should be a priority.

**Accept. We have modified the text accordingly (Lines 114-116) . Thank you.**

Intensity variations and 1d and 2D clustering can occur in the absence of folding (see the 2018 J Struct. Geol. issue on fracture spatial arrangement and Correa et al., 2022, J. Struct. Geol.). So I'm concerned with the impression that the variability observed in this example is necessarily fold related.

**We acknowledge that fracture intensity variations can occur in the absence of folding.  We believe that our fracture intensity maps (Figs. 6, 9, and 10) are compelling, however, and provide a strong case for fold-related fracturing. We are careful to not overstate this point in the manuscript, and by considering intensity variations of the different fracture sets, we show that only hinge-parallel and hinge-perpendicular fractures show systematic changes in fracture intensity with proximity to the fold hinge. We do not claim that all of the observed fractures are fold-related  - in fact, we argue that based on out data, only certain fractures show evidence for being related to folding.**

The Discussion and Conclusions could use reorganization and both sections can be more compact.

**Accept. Thanks for this general comment and the detailed notes below – these have been very helpful. Specific comments are addressed below.**

**DETAILED COMMENTS**

43 There is another major impediment that is worth mentioning because it is in some ways more fundamental than variability and complexity or sampling. This is the inherent lack of inherent complexity in individual fractures and fracture patterns that makes it very challenging to say when and why fractures formed even with good sampling. This is issue of equifinality and the research directions needed to overcome it is described in a recent review: Laubach et al. 2019, Reviews of Geophysics. This issue is fundamental to what this MS is trying to accomplish. Worth addressing explicitly.

Laubach, S.E., Lander, R.H., Criscenti, L.J., et al., 2019. The role of chemistry in fracture pattern development and opportunities to advance interpretations of geological materials. Reviews of Geophysics, 57 (3), 1065-1111. doi:10.1029/2019RG000671

**Accept. We have added some text to lines 66-76 to address this point.**

56-65 Some waning comments about the pitfalls and ways that outcrop fractures can be misleading could perhaps be added here. For using outcrop data to assess specific locations in the subsurface requires more than good exposures and imaging.

**Accept. We have added some text to lines 66-76 to acknowledge that not all fractures observed at outcrop are necessarily representative of the subsurface.**

71 'kinematically consistent'?

**Accept. Text modified accordingly.**

96-102 Where are the specific claims?

**Accept. Text modified accordingly. See Lines 114-116.**

164 Although it probably is a good place to study fold-related fracturing, it seems like demonstrating that these (or some of these) fractures are related to folding is a key point you need to demonstrate. So this statement sounds a bit like the start of a circular argument. Consider revising to clarify your argument.

**Accept. Text modified accordingly.**

177 Consider writing 'Twenty-two' etc as first word in sentence.

**Accept. Text modified accordingly.**

193 (end of methods section) So, no petrology or microstructure of fractures? Seems like a gap in the analysis protocol.

**Some petrology analysis is provided by Watkins et al. (2019). We do not repeat this analysis here. The primary focus of this study is the extraction and analysis of fracture attributes from remote sensing data, with implications for extrapolating fracture properties from field observations to the reservoir scale. As such, detailed microstructural analysis, U-Pb dating etc. is beyond the scope of this work. This would be a great topic for future investigations but we currently don't have sufficient data/samples for this type of analysis. See Lines 228-231.**

In many Paleozoic beds in the Rockies, the fractures are calcite filled, and these deposits can be informative. See for example: Amrouch, K., Lacombe, O., Bellahsen, N., Daniel, J. M., & Callot, J. P. (2010). Stress and strain patterns, kinematics and deformation mechanisms in a basement-cored anticline: Sheep Mountain Anticline, Wyoming. Tectonics, 29(1). And Beaudoin, N., Leprêtre, R., Bellahsen, N., Lacombe, O., Amrouch, K., Callot, J. P., ... & Daniel, J. M. (2012). Structural and microstructural evolution of the Rattlesnake Mountain Anticline (Wyoming, USA): new insights into the Sevier and Laramide orogenic stress build-up in the Bighorn Basin. Tectonophysics, 576, 20-45.

**Thanks for the references. As noted above, we would like to do some more work at Swift focused on cements, chemistry, fluid inclusions, U-Pb etc. This would be a great topic for future research. Given the focus of this paper, this is beyond the scope of the current analysis.**

Fractures formed in the subsurface are in chemically reactive environments where cement deposits are to be expected. So inspecting outcrops for mineral deposits seems like it ought to be step in verification that the sampled outcrop fractures are indeed related to subsurface folding. After all, these rocks are in surface settings where a range of loading conditions could produce fractures. Also, with respect to scaling of open fractures, the diagenesis can set a scale that is worth noting. See Forstner, S.R, Laubach, S.E., 2022. Scale-dependent fracture networks. Journal of Structural Geology, 165, 104748. https://doi.org/10.1016/j.jsg.2022.104748, which is another Rockies example.

**Again, thanks for the reference. I skimmed through the paper – looks like a nice piece of work!**

213 'thrusting' is a process. Consider rephrasing.

**Accept. Text modified accordingly.**

219 The 'outer arc' extension fractures seems like a premature interpretation. What's the evidence for this?

**Accept. Text and Fig. 5 modified accordingly.**

265-275 The patchy and possibly clustered arrangement of some of these fractures is interesting. Clustered backlimb fractures in another Rockies anticline is described by Wang et al. 2023, Quantitative characterization of fracture spatial arrangement and intensity in a reservoir anticline using horizontal wellbore image logs and an outcrop analog. Marine & Petroleum Geology 152, 106238. https://doi.org/10.1016/j.marpetgeo.2023.106238. The analysis methods used there may be of interest.

**Interesting! Fig. 7 in Wang et al. (2023) seems to show that clustering occurs in all structural positions – and no systematic increase in fracture density from backlimb to hinge to forelimb. I can imagine these fractures would show very complex map patterns too. Thanks for the reference – I managed to get it into the Discussion (Lines 623-625).**

290 Will you come back in the discussion to assess the biases in measured lengths? Lots of time the higher resolution the image, the shorter the lengths owing to the segmentation of opening-mode fractures that are increasingly hard to see with standoff.

**Accept. Text added to Lines 306-307**

340 (section) Do you take fractured layer thickness into account in this? Seems like this could be an element in map-view intensity patterns if you have something approaching a fracture / bed thickness relationship. See also 470. How important is it to have omitted this consideration?

**Our analyses did not provide any strong evidence for a relationship between bed (or mechanical layer) thickness and fracture spacing at Swift anticline (Fig. 1, this document). While we acknowledge that this relationship has been documented by numerous previous workers at other localities and that it may potentially influence fracture patterns at Swift anticline, we simply did not observe any compelling evidence for this here. As noted in the manuscript, we find that finer-grained units have higher fracture intensities. Presumably grain size relates to mechanical properties, and in this case, mechanical layer thickness does not appear to play a dominant role. As noted in the revised manuscript (Lines 512-517), it is possible that our bed thicknesses (Fig 5A in the manuscript, Fig. 1 in this document) do not accurately represent mechanical layer thicknesses. We did not assess this in detail during fieldwork and the digital outcrop resolution prohibits unequivocal determination of mechanical layer thickness in this case. It is also possible that we overestimate bed thickness at the site because we do not consider internal laminations and partings within assigned units, hence the poor correlation coefficients below. Future studies could focus on collecting data such as Schmidt rebound measurements, fracture heights (i.e., strata-bound vs. non-strata-bound ), and observations of bed boundaries. Recent work by Bowness et al. (2022), for example, shows that mineralogy exerts a much stronger control on fracture spacing than mechanical layer thickness. We acknowledge your point about this relationship but contend that other factors may be much more important for fracture development here.**

[Figure]

**Figure 1. Fracture intensity vs. bed thickness for the five mapped units (S1-S5). Left column shows correlations between minimum, maximum, and average fracture intensity for all data. Right column shows average fracture intensity for mapped units vs. bed thickness, separated by data type (field data, digital outcrop, and satellite image). Colored lines show different fits (power, logarithmic, and linear) to data.**

393 (section) 'Scaling of fracture orientations' seems like a strange way to say 'impact of observation scale on apparent fracture orientation'.

**Accept. Good suggestion. Section title modified accordingly.**

400 These are interesting graphical representations and maybe should be noted as a feature of the MS in the Introduction.

**Thanks for the note – glad you like them. After careful consideration, we decided to not mention these in the (already too long) Introduction.**

411 But fractures in different structural positions may show differences in *actual* orientation dispersion in different structural positions. For example, Wang et al. 2023 Mar. & Pet. Geol.) in horizontal well data over a Rockies fold, show more orientation dispersion in the steep limb, ascribing in to shear of pre-existing fractures (wing crack formation). I'm not sure I follow your point here. May need clarification.

**Accept. Yes – this is a bit confusing. Sentence deleted.**

449 So would you expect intensity to scale? Why? Is this taken up in the discussion. What to these (seemingly) very different exponents signify?

**Accept. This is addressed in the Discussion (Lines**

420 (figure) so have these orientations/rose diagrams been restored to a pre-folding configuration (has a fold test been done?). How would you do this with just satellite data?

**We conducted fracture mapping in 2D for satellite imagery and in 3D for the digital outcrop (by 3D polyline interpretation in in MOVE). 3D polylines from digital outcrop mapping were projected onto a horizontal plane for orientation and intensity analysis. While this approach does not correct for bed dip and the effects of orientation and intensity distortion, it allows 2D satellite and 3D digital outcrop interpretations to be directly compared within equivalent reference frames. We do not account for orientation distortions because we do not have a reliable method (using the remote sensing approach) for estimating the 3D orientation of fracture traces. We could have rotated fracture traces using bedding orientations based on the assumption that all fractures are perpendicular to bedding. This assumption is not valid at Swift anticline, however, because of the variable dips of fractures with respect to bedding (Fig. 5B in the manuscript). Field images were interpreted in 2D with images oriented according to bed dip at each field station (i.e., images were rotated and scaled so that they had the same orientation as bedding). Fracture interpretations were projected to a horizontal plane for intensity and orientation analysis. Again, while this may introduce some minor geometric artefacts, this approach was taken so that consistency between datasets could be maintained. Text added to lines 214-228.**

460 Ortega et al. (2006, AAPG Bulletin) make the case that you need to specify a scale to have a meaningful, comparable metric of intensity. Is this what you are getting at here?

**Accept. Text modified for clarity.**

Is the 'intensity' here for all sets in aggregate? Even genetically unrelated sets?

**Yes, this is for aggregate intensity. This sentence is used as an introduction to the paragraph before delving into the details of which fracture sets are actually contributing to this increase in intensity and which are not.**

470 Bed thickness effects. This is such a well attested effect on average spacing that it seems strange to write it off like this. With your field data you should be able to at least say what the range of fractured bed thicknesses are and what the height distributions looks like. You cite Hooker

et al. 2013, which has a useful fracture height classification. (Although in my experience with fractured Paleozoic carbonates of the western US, they commonly are not well described by a fractured layer/bed thickness relation, being tip bounded in the classification of Hooker et al. within many meters thick fractured units with narrow spacing). It would be good to have some assessment in the results for your outcrop.

**Thanks for this comment. See detailed response and associated figure above. We agree that the text was a bit dismissive and have modified accordingly. See lines 511-516.**

494 (section) So is the intensity variation with image resolution a consequence of scaling of fracture sizes or is it an artifact of image resolutions? Do you mean the *average orientation* depending on the volume of rock over which you aggregate measurements? This seems confusing. For fracture sets having power law size distribution (as in the Hooker et al. 2009 reference noted above) fractures in the same set may share *common* (identical) orientations over orders of magnitude in fracture size. But this wouldn't mean that the orientations 'scale' would it?

**Accept. We agree that this paragraph is a bit confusing and probably best removed, given the difficulty in distinguishing between actual fracture scaling and the effects of image resolution from our data. Paragraph deleted.**

495-500 But the prominent title of the paper is 'scaling'. Does this mean the scaling is not so important? Clarify.

**Accept. Sentence deleted.**

500-505 Hmm. This seems a bit inconclusive.

**Accept. Sentence deleted.**

510 Although really, the Stearns and Price models have long been questioned including the hkl fractures in Hancock (1985) and more specifically for the Rockies by Hennings et al. 2000. Does anyone really still rely on the Stearns model? (the hkl scheme also captures 'variable and dispersed' although that ends up a criticism of such an 'expanded' Stearns model, since every orientation can be fit in.)

**Accept – this is a good point. Text modified accordingly.**

A related point is that for angular-hinged folds with rotated but otherwise undeformed limbs, the fold-related fracture strain can be quite localized in a narrow (and frequently poorly exposed) hinge region.

**Yes – this likely depends on the type of fold and how/when the fractures formed. Our fracture intensity maps, we think, provide a compelling case for our set A and set B fractures being fold related over a relatively wide zone.**

512-513 and to 518. These arguments do not seems overly convincing. (i) Is the 'in general' increase in intensity toward the fold hinge quantitatively different from intensity variations in backlimb setting distant from hinges? Can you even test this with the degree of exposure? (ii) The 'parallel' and 'perpendicular' to the fold hinge configuration are also orientations that are in kinematic compatibility with other regional and local features, including former basin margins (as postulated for some other Rockies fractures by Lorenz in the '90s), the Cordilleran extensional province boundary and many young normal faults in the norther Rockies, and local topography. In some Rockies folds these seemingly kinematically compatible orientations are fully or in part out of sync with the timing of folding (Laubach, S.E., Fall, A., Copley, L.K., Marrett, R., Wilkins, S., 2016. Fracture porosity creation and persistence in a basement-involved Laramide fold, Upper Cretaceous Frontier Formation, Green River Basin, U.S.A. Geological Magazine 153 (5/6), 887-910. doi:10.1017/S0016756816000157).

**We would argue that the fracture intensity maps (Figs. 6, 9, and 10) show a compelling case for increased fracture intensity towards the hinge of the fold, with backlimb positions showing lower intensities, particularly for the hinge-parallel and hinge-perpendicular sets. We believe that hinge, forelimb, and backlimb positions are exposed sufficiently to assess this relationship and therefore that yes, we can test this with the degree of exposure. We take your point about the orientations of fractures 'parallel' and 'perpendicular' to the fold hinge being in kinematic compatibility with other regional and local features. This doesn't explain the clear increase in abundance of these features towards the fold hinge, however. Based on the clear increase of the parallel and perpendicular sets towards the fold hinge, we see no reason to invoke other regional or local tectonic features such as basin margins, extensional province boundaries etc., particularly as we see no compelling evidence for these other features.**

Is there any crossing and abutting relations evidence to see if the two sets formed in the expected sequence?

**As noted in the manuscript, abutting or cross-cutting relationships are inconclusive at the site and we therefore do not speculate on pre and post-folding fracture timing. We make the observation that sets A and B increase in intensity towards the fold hinge and are therefore (in the absence of any contradictory evidence) related to fold formation. The other fracture sets do not show this relationship and are therefore not interpreted as related to fold formation. We do not have absolute ages of the fracture sets and therefore do not speculate on their timings.**

519 The passive construction here obscures who it is doing the interpreting ('are generally interpreted to...) If this is your interpretation, construct the sentence to make this clear then say why you think so. Why could these not be (at least in part) pre- or post-folding regional fractures? (Regional fractures having about this orientation are present elsewhere in the Rockies distant from fold hinges, and some of these arrays are clustered).

**Accept. We have changed the sentence construction as suggested. We acknowledge that some of these structures may be related to other deformation events but, as stated above, the clear increase in fracture intensity towards the hinge suggests that sets A and B are likely fold related.**

527 I think this process was explicitly postulated for fractures in the Sevier fold belt by Lorenz. But may be published in an obscure source.

**Thanks. We dug around for this reference but couldn't find the paper you are referring to.**

530 Hmm. Maybe you should say this first and then provide the speculation.

**Accept. Text modified accordingly.**

540-545 But these caveats also apply to the fracture sets that happen to coincide with your fold geometry. Maybe it would be better to provide this broader context first then make the case for linking some fractures to the fold.

**We believe, as stated previously, that the increases in fracture sets A and B intensity towards the fold hinge are compelling. This site has likely experienced multiple episodes of deformation but this doesn't explain the increased intensity of sets A and B towards the fold hinge. It does, however, help to explain the variable orientations and complex patterns observed at the site.**

Overall, I think the Discussion can be more compact.

**Accept. We have tried to trim it down where possible. Thanks.**

575 (figure) is there a reason that the fractures shown don't have any particular crossing and abutting relations? The text makes it seem like the fold contains temporally and genetically distinct groups of fractures and wouldn't you expect such patterns from a fold-related progression of fracture? Especially if (as is likely) preexisting fractures exist in the folded layers?

**The reason we don't include explicitly account for cross-cutting or abutting relationships in this figure is that, as noted above, cross-cutting and abutting relationships at the site are inconclusive. As noted above, rather than speculate on fracture timing we describe which fractures are likely fold related and which are not. Taking a position on this would require further analysis, including, if possible, absolute dating of fracture fill.  Given that our focus is on remote sensing analysis, this is beyond the scope of the current work.**

591 Is this really the appropriate topic sentence for this conclusion? Some hinge-parallel and hinge-perpendicular fractures systematically increase in abundance towards the fold hinge and are thus presumably fold related. But many fractures/sets are not clearly related to folding and you say they are probably unrelated to folding. The composite pattern from the outcrop is some fold-related fractures amidst others of unknown origin.

**Accept. Text modified accordingly.**

580-90 The 'are attributed' construction is not helpful. Instead say 'we attribute' and then say why (what about the 'stratigraphic level'?). Your claim here is not clear. The highest intensity would be in the fold hinge, but some units have higher overall intensity because of rock type or bed thickness

and where these are exposed away from the fold hinge it results in a discrepancy between intensity and structural position?

**Accept. Text modified accordingly.**

There are some formatting corrections needed in the reference list.

**Accept. Thanks. Formatting corrections addressed.**

**References**

**Bowness, N.P., Cawood, A.J., Ferrill, D.A., Smart, K.J. and Bellow, H.B., 2022. Mineralogy controls fracture containment in mechanically layered carbonates. Geological Magazine, 159(11-12), pp.1855-1873.**

---

## Author Comment (AC2)

**REVIEWER 2 (CORRADETTI): COMMENTS AND RESPONSES**

Reviewer comments in black, responses in blue. Where line numbers are specified in responses, these refer to the "track changes" version of the revised manuscript.

**GENERAL COMMENTS**

The manuscript by Adam Cawood and coauthors presents a remote-sensing study investigating the natural fracture patterns at Swift anticline in NW Montana. The study utilizes three datasets: satellite images, drone-derived SfM models, and field investigations with accompanying oriented photographs. According to the authors, this progression of datasets provides observations at low, medium, and high resolutions, respectively. It should be noted that one of the datasets presented has been previously published by the same authors, which is acknowledged in the manuscript.

Thank you for the detailed and thorough review. Note, the only data reproduced from Watkins et al. (2019) are the hinge trace position (e.g., Fig. 4, this manuscript) and the estimated fracture intensity, which we compare to fracture intensity derived from remote sensing data (Fig. 14, this manuscript). Fracture orientations derived from field data (e.g., Fig. 12 this manuscript) were not reported in Watkins et al. (2019) and are therefore original data for this study.

In general, the manuscript falls within the scope of the SE scientific field, and while the concepts presented are not entirely novel, they hold value for further exploration. The manuscript is well-written and exhibits clear organization. However, there are aspects of the methodology and data presentation that require additional clarification. For instance, it would be beneficial for the authors to elaborate on how the hinge trace was mapped, such as whether a 3D curvature map or bedding information was utilized.

Thank you for your comments. Your comment regarding the hinge trace is addressed in the responses to detailed comments below.

To improve clarity, I suggest including an early cautionary note in the manuscript addressing the limitations posed by vegetation cover and erosion at the study site.

Accept. We have added some text to lines 70-72 to acknowledge that vegetation cover and erosion may have impacted our results. Note that these factors are likely to influence fracture mapping at many outcrops and therefore Swift anticline is not unique in this respect.

Additionally, the explanation of fracture map analysis would benefit from further details, particularly regarding the consideration of bedding orientation.

Accept. Addressed in detailed comments below.

If available, providing a description and chronology identification of fracture types observed in the field would enhance the manuscript.

As noted in lines 102-108, the primary focus of this study is the extraction and analysis of fracture attributes from remote sensing data, with implications for extrapolating fracture properties across observation scales. As such, detailed, field-scale observations of fracture morphology (e.g., fracture cements and kinematic indicators) are beyond the scope of this work. While these types of

**observations are of course important, our rationale for not including these data here are that we focus here on comparing field observations with results derived from remotely acquired data.**

Below is a list of specific comments on individual lines of the manuscript, I hope the authors will find them useful and constructive.

**Thank you very much. Your comments have been very insightful, constructive, and helpful. Detailed comments and questions are addressed below – we hope you find our responses satisfactory.**

**DETAILED COMMENTS**

Line 16: Can we define a fault-related fold as a structurally complex setting, or are there additional factors to consider at the outcrop?

**Accept. We have modified the text accordingly. Thank you.**

Line 21: Regarding point ii, I apologize if I am misunderstanding, but it seems to suggest that any observations made approximately parallel or perpendicular to the hinge are indeed parallel or perpendicular to the hinge. Perhaps there is a clearer way to convey this, especially considering that fracture types and their intersections were not characterized in the field.

**Accept. This is a good point, thank you. We have modified the text for clarity (Line 23).**

Line 96: Considering the well-known relationship between fracture spacing and mechanical unit thickness, particularly in granular rocks, I wonder if any of these other parameters can be correlated without taking into account the mechanical unit thicknesses.

**Our analyses did not provide any strong evidence for a relationship between bed (or mechanical layer) thickness and fracture spacing at Swift anticline (Fig. 1, this document). While we acknowledge that this relationship has been documented by numerous previous workers at other localities and that it may potentially influence fracture patterns at Swift anticline, we simply did not observe any compelling evidence for this here. As noted in the manuscript, we find that finer-grained units have higher fracture intensities. Presumably grain size relates to mechanical properties, and in this case, mechanical layer thickness does not appear to play a dominant role. As noted in the revised manuscript (512-518), it is possible that our bed thicknesses (Fig 5A in the manuscript, Fig. 1 in this document) do not accurately represent mechanical layer thicknesses. We did not assess this in detail during fieldwork and the digital outcrop resolution prohibits unequivocal determination of mechanical layer thickness in this case. It is possible that we overestimate bed thickness at the site because we do not consider internal laminations and partings within assigned units. Future studies could focus on collecting data such as Schmidt rebound measurements, fracture heights (i.e., strata-bound vs. non-strata-bound ), and observations of bed boundaries. Despite these limitations, we contend that other factors can be considered without the need for explicitly accounting for bed thickness. Recent work by Bowness et al. (2022), for example, shows that mineralogy exerts a much stronger control on fracture spacing than mechanical layer thickness. We acknowledge your point about this relationship but contend that other factors may be much more important for fracture development.**

[Figure]

**Figure 1. Fracture intensity vs. bed thickness for the five mapped units (S1-S5). Left column shows correlations between minimum, maximum, and average fracture intensity for all data. Right column shows average fracture intensity for mapped units vs. bed thickness, separated by data type (field data, digital outcrop, and satellite image). Colored lines show different fits (power, logarithmic, and linear) to data.**

Figure 4: While I have not personally visited this anticline, based on the provided figure, particularly panels A and C, it appears that the exposed forelimb is limited, potentially due to erosion or vegetation cover. This prompts questions about how the hinge zone was delineated. Since a detailed 3D model of the fold is available, obtaining the curvature of the structure should be relatively straightforward. I suggest delving into this aspect further.

**Thank you for the comment. Watkins et al. (2019) constructed a 3D surface of the top Castle Reef Formation from bedding data and field observations (see Fig. 5; Watkins et al., 2019). This constructed surface was used in part by Watkins et al. (2019) to define the hinge position. As you note, much of**

**the forelimb is missing at Swift anticline and therefore precisely defining the hinge position is not straightforward. Further, the stepped erosional profile and vegetation at the site make construction of a "clean" 3D model challenging. For consistency, we have used the hinge line of Watkins et al. (2019). We acknowledge that the position of this hinge line could be different from what is shown here and in Watkins et al. (2019) but because of the missing forelimb exposure, any estimation of the hinge position would have to be generated through construction techniques and this would be subject to uncertainties. We have added text (Lines 189-191) to clarify that the Watkins et al. (2019) hinge position was used in this study.**

Lines 166: Please clarify what is meant by "characterizing overall structural geometries." The manuscript would have benefitted from a description of fracture types and their chronological identification in the field.

**Accept. We have modified the text for clarity. As noted above, the primary focus of this study is the use of remote sensing data for fracture characterization. As such, detailed field-based observations (e.g., documentation of kinematic indicators) or laboratory analyses (e.g., U-Pb dating of fracture filling calcite) were not the primary focus of this study. We take your point about fracture timing etc. but this paper is essentially focused on the use of remote sensing data for fracture analysis – as such, we limited our analyses to observations of fracture patterns at multiple scales.**

Lines 173-183: As a point of discussion, it is evident that the ground pixel resolution of these two datasets is insufficient to represent a change in scale; in fact, they are at the same scale.

**While these data have pixel resolutions within the same order of magnitude, they have different scales if scale is considered as a graduated range of values.**

I wonder if the difference in mapped fracture orientation can be attributed to variations in vegetation cover (assuming the datasets were acquired in different years or seasons) or differences in mapping strategies. For example, it is conceivable that the satellite photoset was analyzed solely in a vertical (nadir) direction, whereas the 3D model could have incorporated bedding attitude, thus always being perpendicular to the bedding. In this context, it remains unclear whether the bedding attitude was considered as it should be. Oblique fracture traces in relation to the hinge may exhibit different orientations when observed obliquely to the bed-perpendicular direction. Please provide a clearer explanation of how the data were handled.

**This is a good point. In response to your concern regarding variations in vegetation, our images were collected in June 2016 and the Google Earth imagery is dated from July 2014. Based on the similar times of year that our data and the satellite imagery were collected, it is unlikely that major seasonal variations can account for changes in vegetation between the datasets. It is possible that more vegetation may be present at the site in our imagery than in the satellite data due to vegetation growth over two years (we saw no evidence for recent wildfires at the site so the converse is unlikely). This doesn't explain the lower fracture intensities from satellite imagery, however – if anything we might expect fewer fractures to be observed after several years of vegetation growth (i.e., in the digital outcrop data). Further, based on visual comparisons between satellite imagery (Fig. 4A in the manuscript) and our data (Fig. 4B and: https://sketchfab.com/3d-models/swift-anticline-montana-4c60c376a2984166843fc3391b2a85b7) there do not appear to be significant differences in vegetation patterns between July 2014 and June 2016.**

**Regarding orientations (again this is a good point), we conducted fracture mapping in 2D for satellite imagery and in 3D for the digital outcrop (by 3D polyline interpretation in in MOVE). 3D polylines from digital outcrop mapping were projected onto a horizontal plane for orientation and intensity analysis. While this approach does not correct for bed dip and the effects of orientation and intensity distortion, it allows 2D satellite and 3D digital outcrop interpretations to be directly compared within equivalent reference frames. There is no straightforward way to account for geometric artefacts in satellite imagery (e.g., steeper beds may appear to have more closely-spaced fractures in the dip direction than is real) and therefore we elected to treat all fracture maps as essentially horizontal. While this may lead to overestimates of fracture intensity in fractures oriented perpendicular to the dip direction, we consider this effect to be relatively minor at the scale of the analysis area. Note that most of the pavements exposed on the crest of Swift anticline have dips around 20$^O$ or less and therefore we expect the effects of intensity distortion to be relatively minor. We do not account for orientation distortions because we do not have a reliable method (using the remote sensing approach) for estimating the 3D orientation of fracture traces. We could have rotated fracture traces using bedding orientations based on the assumption that all fractures are perpendicular to bedding. This assumption is not valid at Swift anticline, however, because of the variable dips of fractures with respect to bedding (Fig. 5B in the manuscript). Field images were interpreted in 2D with images oriented according to bed dip at each field station (i.e., images were rotated and scaled so that they had the same orientation as bedding). Fracture interpretations were projected to a horizontal plane for intensity and orientation analysis. Again, while this may introduce some minor geometric artefacts, this approach was taken so that consistency between datasets could be maintained. Text added to lines 211-225.**

Lines 189-190: Once again, it would be beneficial to provide more information regarding the orientation of digitization in relation to the bedding attitude and the analysis of fracture traces (which may vary accordingly).

**Accept. See details above and Lines 213-227 in the revised manuscript.**

Figure 6B: It appears that vegetation significantly covers large portions of the fold. Therefore, using an overall map that compares vegetated areas with bare areas might not be the most suitable approach for assessing variations in fracture intensity. I suggest adding a cautionary note to acknowledge this limitation in the dataset if the authors, who are familiar with the exposure, believe that vegetation could be a factor affecting the data.

**Accept. We have added text to Lines 70-74 to acknowledge this. As noted above, these factors are likely to influence fracture mapping at many outcrops and therefore Swift anticline is not unique in this respect.**

Lines 301-302: I followed the link to the low-resolution model and found that its availability does not contribute significantly to the understanding of the area. Although I understand the size restrictions on Sketchfab, it may be worthwhile to improve the texture map, which is crucial in this context, to allow readers to visualize some of the mapped fractures. Additionally, why was a link provider used instead of providing a direct link to Sketchfab?

**Unfortunately, despite multiple attempts, we have not been able to upload a clearer image of the study site via Sketchfab. As you note, size restrictions on Sketchfab are often prohibitively small,**

particularly for outcrops of this size. The primary justification for providing this link is so that readers can get a feeling for the 3D nature of the site, rather than being able to observe all fractures in detail. As suggested, we have replaced the shortened URL with the native link.

Lines 333-334: Please specify if these discontinuous patches are clearly separated by covered portions of the anticline.

This "patchiness" is not because of vegetated areas on the anticline crest but rather because of change in fracture intensity on exposed bedding surfaces. (see Figs. 4 and 6 in the manuscript).

Line 336: How was the hinge position identified? Was the curvature of the 3D model used?

Accept. As noted above we use the hinge line of Watkins et al. (2019). See Lines 189-191 in the revised manuscript.

Lines 395-397: It may be repetitive, as I previously mentioned in the Methods section, but I am curious about how the fracture maps were analyzed. I presume field photographs were always acquired perpendicular to the bedding; was this consideration taken into account when comparing orientation data from satellite images? If not, please clarify for better understanding.

Accept. See detailed response above and Lines 213-228 in the revised manuscript.

Line 525: It would be helpful to present these observations within the data presentation section.

Accept. We have added this information to the results section on Lines 294.

Line 529: This is the first instance where difficulties in accurately estimating the fold curvature and, consequently, the fold hinge are mentioned. It would be more appropriate to introduce some of these aspects earlier in the manuscript. Based on my understanding, although I am not familiar with the site, it seems that the maximum curvature of the fold may be located further east than its interpreted position.

Accept. As noted above, we use the hinge line of Watkins et al. (2019) for consistency. We have added to text to Lines 189-191 to make this point.

Lines 531-543: Consistent with the initial stages of layer-parallel shortening, conjugate sets of strike-slip faults may have formed, aligning with the observed orientations at Swift Anticline. For example, refer to this review paper (10.1016/j.earscirev.2014.11.013) and the references therein. Since the position of the maximum curvature is speculative and may exhibit local variations, as demonstrated in this study, some of these earlier structures should be compared with the regional shortening direction rather than solely with the hinge trace.

Accept. Text added to include this point. Lines 593-605. Again, this study is focused on mapping fractures using remote sensing data. We absolutely acknowledge that conjugate strike slip faults may have formed at this site during the early phases of contraction, but given that we do not have unequivocal data regarding this, we have decided not to include this interpretation in the study. As we note above and in the manuscript, we do not have sufficient data to determine the timing of fractures at the site. Future studies could focus on this aspect.

Line 568: Should it be "during folding" instead of "after"?

Accept. Text modified accordingly. Thanks.

**References**

Bowness, N.P., Cawood, A.J., Ferrill, D.A., Smart, K.J. and Bellow, H.B., 2022. Mineralogy controls fracture containment in mechanically layered carbonates. Geological Magazine, 159(11-12), pp.1855-1873.